# An archaeal nucleoid-associated protein binds an essential motif in DNA replication origins

Rajkumar Dhanaraju[1], Rachel Y. Samson[1,2,3,4], Xu Feng [1,5], Alessandro Costa [3,6], Giovanni Gonzalez-Gutierrez [1] & Stephen D. Bell [1,2,3,4] ✉

DNA replication typically has defined start sites, or replication origins, which are designated by their recognition by specific initiator proteins. In addition to initiators, general chromatin or nucleoid-associated proteins have been shown to play roles in modulating origin efficiency in eukaryotes and bacteria. The role of chromatin proteins in origin function in the archaeal domain of life is poorly understood. Here, we describe a dissection of sequences elements required for in vivo function of an archaeal DNA replication origin. Our data reveal a hitherto uncharacterized sequence element, the *ucm*, is required for origin activity. We identify a protein, UBP, that interacts with the *ucm* and additionally with hundreds of other sites on the genome. We solve the crystal structure of UBP alone and in complex with *ucm* DNA, and further show that UBP interacts with the MCM replicative helicase. Taken together, our data provide evidence that UBP functions as a general nucleoid-associated protein that plays a key role in facilitating the egress of the MCM replicative helicase from DNA replication origins.

First articulated over 60 years ago, the replicon hypothesis posits that DNA replication initiation is dependent on two elements: a replicator element that acts in *cis* as a site, or origin, of DNA replication initiation and a structural gene that encodes a *trans*-acting protein that interacts with the replicator element and leads to replication initiation[1]. The validity of this insightful theory has been supported by numerous studies in bacteria, leading to the identification and characterization of *cis*-acting DNA replication origins that are acted on by the highly conserved bacterial replication initiator protein, DnaA[2]. The general applicability of the hypothesis to eukaryotic organisms appears to be more variable across the phyletic diversity of eukaryotes. Well-defined *cis*- and *trans*-acting elements have been extensively characterized in budding yeast[3]. However, even in yeast with its sequence-defined

replication origins, nucleosome positioning has a profound influence on origin activity[4]. Furthermore, organisms in more recently branching phyla of eukarya show increasing reliance on more complex chromatin structure-based and epigenetic cues to conditionally define replication initiation sites[4–6]. Significantly, even in bacteria, DNA-binding nucleoid-associated proteins (NAPs) such IHF and Fis have been long-known to play key roles in modulating the function of the *Escherichia coli* DNA replication origin *oriC*[2,7].

In contrast to the extensive body of literature regarding bacterial and eukaryotic DNA replication, comparatively little is known about how replication initiation occurs in archaea[8]. The core archaeal replication machinery is clearly orthologous to that of eukaryotes and could be viewed as representing a simplified and ancestral form of the

[1]Department of Molecular and Cellular Biochemistry, Indiana University, Simon Hall MSB1, 212 S Hawthorne Drive, Bloomington, IN 47405, USA. [2]Department of Biology, Indiana University, Simon Hall MSB1, 212 S Hawthorne Drive, Bloomington, IN 47405, USA. [3]Sir William Dunn School of Pathology, South Parks Road, Oxford OX1 3RE, UK. [4]Present address: Department of Microbiology, The Ohio State University, Aronoff Laboratory 218, 318 W 12th Ave, Columbus, OH 43210, USA. [5]Present address: State Key Laboratory of Microbial Technology, Shandong University, Qingdao 266237, China. [6]Present address: Macromolecular Machines Laboratory, The Francis Crick Institute, London, UK. ✉e-mail: bell.2007@osu.edu

eukaryal machinery[9]. For example, the eukaryotic replicative helicase is based on the heterohexameric Mcm(2–7) complex. All six subunits are related to one another, suggestive of descent from a common ancestor, and the archaeal MCM exists as a homohexamer[10]. In eukaryotes, Mcm(2–7) is recruited to origins by the action of another heterohexameric complex, ORC, that serves as the origin-defining initiator complex. Mcm(2–7) loading at origins requires the activities of two co-loaders, Cdc6 and Cdt1. The Orc1 subunit of ORC and Cdc6 share sequence similarity, and archaeal genomes typically encode one or more proteins related to Orc1 and Cdc6[11]. Significantly, in vitro reconstitution reveals that the ATP-bound form of the archaeal initiator protein Orc1–1 from the species *Sulfolobus islandicus* is necessary and sufficient to recruit MCM to origin DNA, indicating that this archaeal protein possesses both initiator and loader activities of its derived eukaryotic counterparts[12]. In addition, many archaeal chromosomes are replicated from multiple replication origins, in contrast to the single replication sites typically found in bacterial chromosomes and more reminiscent of the situation in eukarya[13]. However, in eukarya, it is generally believed that all origins are bound by the ORC complex[3]. In contrast, genetic studies in the archaeon *S. islandicus* have found that each of the three replication origins (*oriC1*, *oriC2*, and *oriC3*) found in the *Sulfolobus* chromosome are recognized and defined by distinct initiator proteins (Orc1–1, Orc1–3, and WhiP, respectively)[14]. Of the three *Sulfolobus* origins, the Orc1–1-defined *oriC1* is most highly conserved across the diversity of archaea and consequently has been the subject of most research. *Sulfolobus oriC1* has three binding elements for Orc1–1, termed origin recognition boxes (ORBs)[15]. The ORBs each bind a monomer of Orc1–1 and do so with a defined polarity[16]. ORB2 and ORB3 are present in inverted orientation to one another and flank a highly AT-rich region, a potential duplex unwinding element (DUE). Replication initiation point mapping revealed replication starts at the juxtaposition of the AT-rich region and ORB3[15]. Disruption of the ORB3-DUE-ORB2 sequences ablates origin activity in vivo[17]. Interestingly, examination of the sequence of the DUE of *Sulfolobus* (now *Saccharolobus*) *solfataricus* P2 *oriC1* revealed the presence of a short sequence motif that was also found in *oriC2* and *oriC3*. This was well conserved in the *Sulfolobales* as well as in the two replication origins of the more distantly related *Aeroyrum pernix* (a member of the *Desulfurococcales* phylum). This motif, of then unknown function, was termed the *ucm* (**u**n**c**haracterized **m**otif)[18].

In the current work, we describe an analysis of *cis*-sequence requirements for the genetic function of *oriC1* in *S. islandicus* REY15A. We find a critical importance for both ORB2 and ORB3. Loss of either of these elements abrogates origin activity. Additionally, mutation of the *ucm* severely reduced origin activity. We identify a hitherto uncharacterized protein factor that we term UBP (**u**cm-**b**inding **p**rotein) that interacts sequence-specifically with the *ucm* and determine the structure of the protein both in isolation and when bound to DNA. UBP is a highly abundant protein, and we reveal that it plays a role akin to bacterial nucleoid-associated proteins with over 300 binding sites on the *Sulfolobus* chromosome. Importantly, UBP interacts with the N-terminal domains of MCM. However, chromatin immunoprecipitation reveals *ucm* disruption does not affect MCM recruitment to the origin but, rather, results in elevated occupancy of MCM at the locus. These data, in conjunction with biochemical reconstitution experiments, lead us to propose a role for UBP in origin activation.

## Results

### The *ucm* is required for origin activity in vivo

To assess the sequence requirements for origin function we exploited the endogenous CRISPR system of *S. islandicus* REY15A to introduce targeted linker-scanning mutations into the chromosomal *oriC1* locus (Fig. 1a, Supplementary Fig. 1). We introduced 10 bp substitutions across the 130 bp ORB3-DUE-ORB2 region and, by utilizing whole-genome marker frequency analysis (MFA-Seq), assayed the ability of the resultant strains to support replication from the mutated origins (Fig. 1b). MFA-Seq performed on wild-type (WT) cells reveals the typical profile seen in *Sulfolobus* with three peaks, corresponding to replication initiating at, and proceeding bidirectionally from, all three replication origins[14,19,20]. Notably, mutation of either ORB2 or ORB3 (mutants M1, M2, M12, and M13) resulted in loss of the peak at *oriC1* indicating that these motifs are individually required for bidirectional replication. The DUE between ORB2 and ORB3 was largely insensitive to mutation, with the notable exception of the M6 and M7 strains, in which replication from *oriC1* was undetectable. The substitutions in M6 and M7 overlap the *ucm* motif, underscoring the importance of this motif for origin function.

Given that activity of *oriC1* requires the Orc1–1 protein and the *orc1–1* gene is immediately adjacent to the origin, we wished to test whether any of the origin-affecting mutations impacted the expression levels of Orc1–1. As can be seen in Fig. 1c, Orc1–1 was expressed at wild-type levels in all the mutant lines.

### Identification of UBP—a *ucm*-binding protein

Next, we speculated that the *ucm* could serve as a binding site for an as-yet-unidentified protein. We performed sequential ion-exchange, heparin-affinity and DNA affinity chromatography, following purification with electrophoretic mobility shift assays. The final step utilized double-stranded DNA oligonucleotides corresponding either to the wild-type *ucm* sequence or in which the sequence of the *ucm* was mutated (Fig. 2a). The control sequence, with mutated *ucm* (with A-C, G-T, C-A and T-G transversion substitutions in the 11 bp *ucm* sequence), led to the purification of a single protein species that was identified as the Alba protein—a highly-abundant non-specific DNA binding protein in *Sulfolobus*[21,22]. Alba also bound to the *ucm*-containing oligonucleotide, however a second protein that we termed UBP (for **u**cm-**b**inding **p**rotein) was additionally eluted from this affinity matrix. The affinity purification was performed with extracts from *Sulfolobus acidocaldarius* and led to the identification of UBP as the product of the Saci_0847 open reading frame. For subsequent work, we have studied the single ortholog from *S. islandicus* REY15A, a 111 amino acid basic protein (calculated pI = 9.3) encoded by the SiRe_1573 gene. A genome-wide survey of gene essentiality in *S. islandicus* M.16.4 revealed the gene for UBP to be essential for viability[23].

We purified a recombinant version of UBP following over-expression in *E. coli*. Electrophoretic mobility shift assays (EMSAs) revealed the protein bound specifically to double-stranded DNA oligonucleotides containing the *ucm* sequence but not to scrambled-sequence oligos (Fig. 2b). Binding was specific to double-stranded DNA; neither single strand was detectably bound by UBP (Fig. 2c). Additionally, UBP showed greatly reduced binding to 50-nucleotide-length oligonucleotides containing the sequence substitutions introduced in the *oriC1*-inactive M7 mutant strain (Fig. 2d). Yield of the UBP-complex was visibly reduced when binding oligonucleotides containing the M6 substitutions. Intriguingly, quantitation of unbound DNA reveals a similar profile for the M6 substrate to that seen with wild-type sequences. Thus, it appears that the M6 mutations impair the stability of the protein-DNA interaction, resulting in complex dissociation during electrophoresis (Fig. 2d). Finally, interaction of UBP with DNA was able to thermally protect duplex DNA from thermal melting (Fig. 2e).

DNaseI footprinting on *oriC1* sequences revealed UBP generates a pattern of protection centered around on the *ucm* sequence (Fig. 2f). Notably, the inclusion of Orc1-1 in the footprinting assays resulted in a composite of UBP-alone and Orc1–1–alone protection patterns (Fig. 2g). No additional protection or DNaseI hypersensitivity was observed, suggesting that, at least on this linear substrate, there was no interaction between Orc1–1 and UBP on the DNA template.

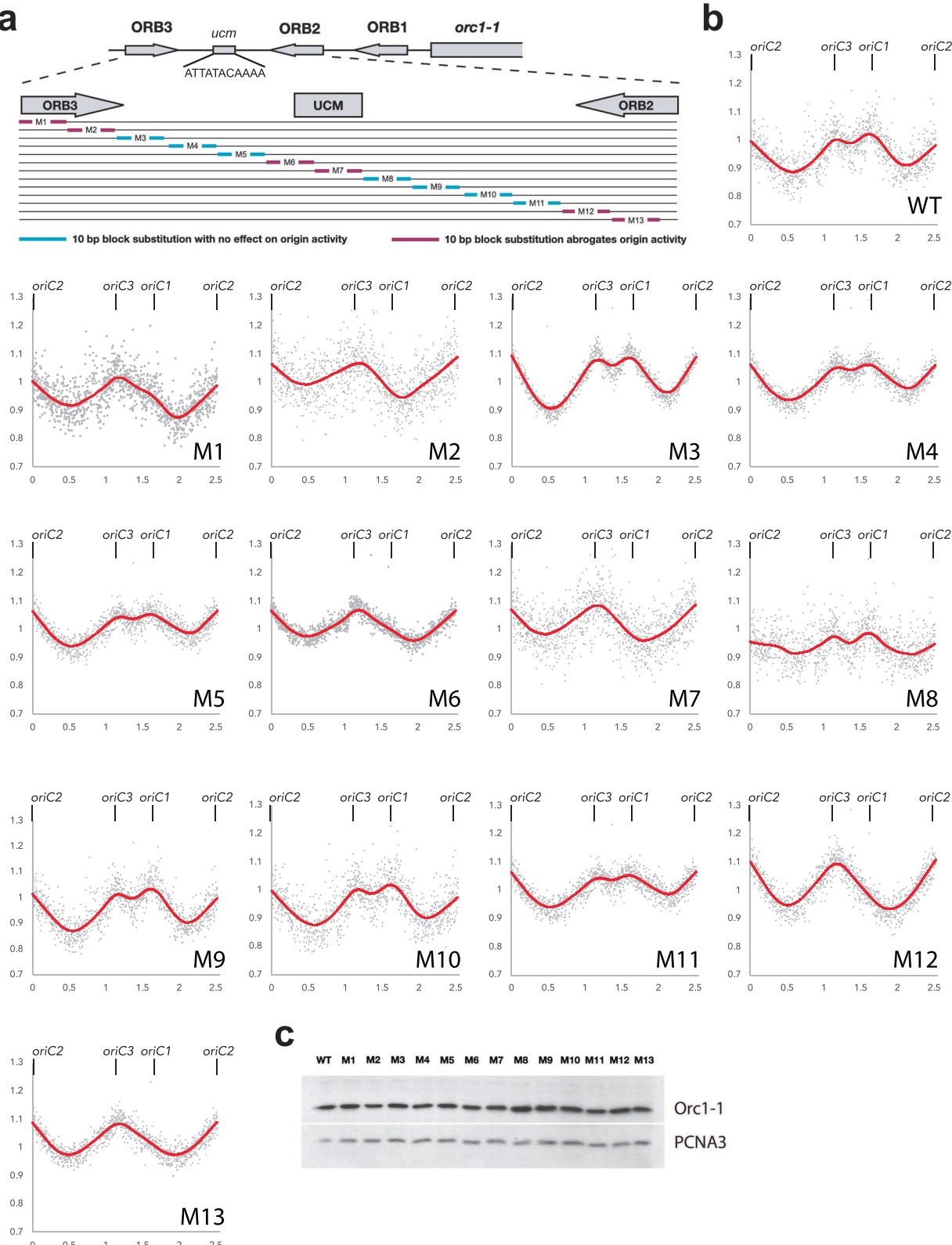

**Fig. 1 | Determination of sequence requirements for *oriC1* function in vivo.**
**a** Schematic of the organization of *Sulfolobus islandicus oriC1*. The upper panel shows the position and orientation of the Orc1-1-binding ORB elements, the *ucm* and the position of the *orc1-1* gene. The expanded panel below indicates the regions targeted in the linker scanning mutagenesis. Substitutions that impact on origin activity are shown in purple while those with no effect are shown in cyan. **b** Marker frequency assays with wild-type (WT) and the indicated mutants (M1–M13) are shown. The position of the three replication origins are shown above each panel (oriC2 lies at the point of in silico linearization of the circular genome and is thus indicated at the beginning and end of the X-axis). Data were binned in 1 kb intervals and normalized to wild-type stationary phase cell DNA content. **c** Western blot analysis of whole cell extracts from wild-type (WT) and the indicated mutants (M1–M13) with antisera generated against Orc1-1 (upper panel) or PCNA3 (lower panel), the latter serving as a loading control.

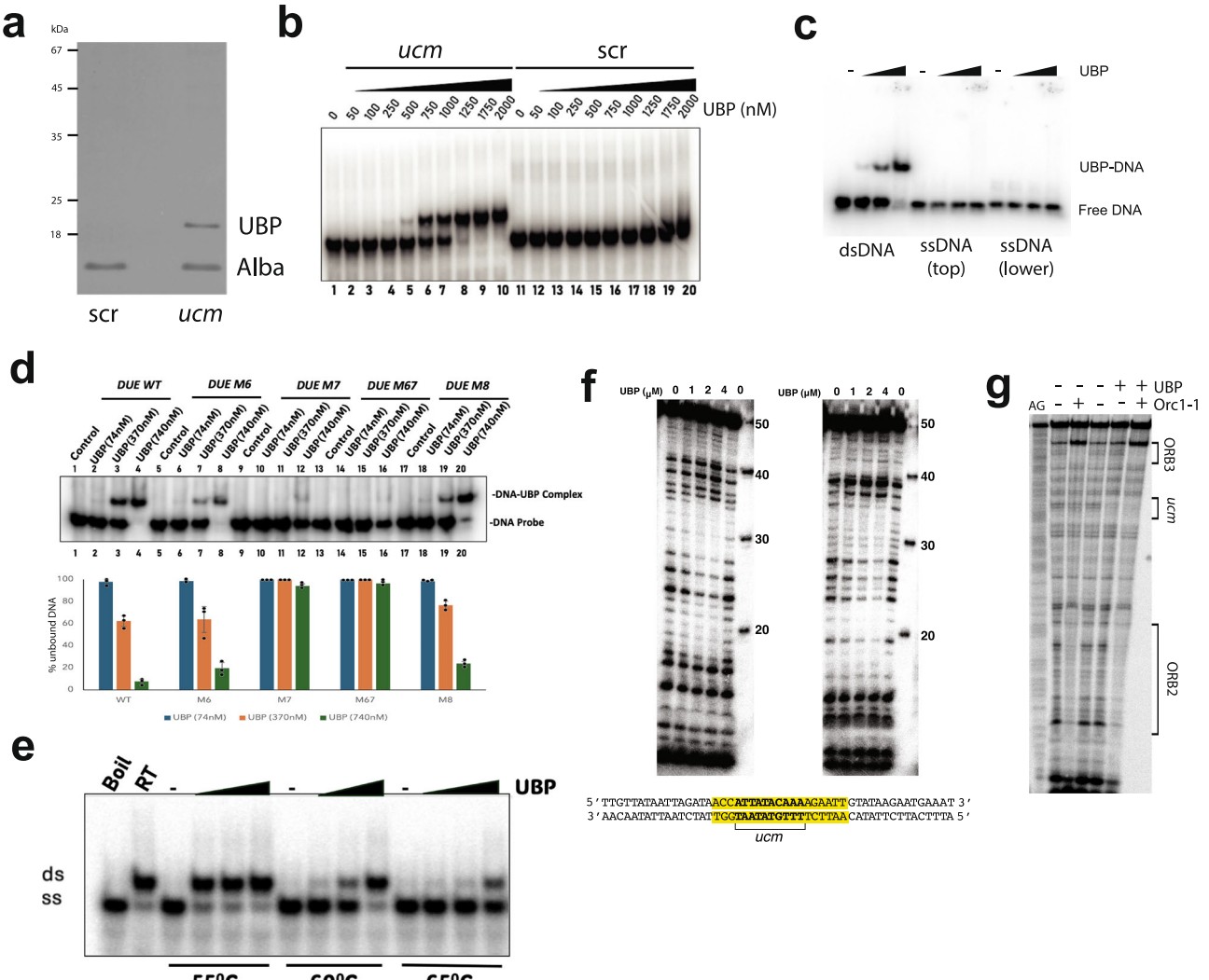

**Fig. 2 | Identification and characterization of UBP. a** SDS PAGE analysis of eluate from DNA affinity columns with *ucm*-containing oligonucleotides (ucm) or oligonucleotides with the same overall base composition but scrambled sequence (scr). Protein identity was confirmed by mass spectrometry by the Sir William Dunn School of Pathology mass spectrometry facility. Note that UBP migrates anomalously slowly on SDS PAGE, presumably a consequence of its basic pI. **b** Electrophoretic mobility shift assays (EMSAs) with the indicated concentrations of purified recombinant UBP and with 50 bp double-stranded DNA oligonucleotides containing ucm (*ucm*) or scrambled (scr) sequence. **c** EMSAs with zero, 74, 370, or 740 nM UBP and 20 bp double-stranded *ucm*-containing oligonucleotides or either top or lower single-stranded oligonucleotides. **d** EMSAs with 50 bp double-stranded oligonucleotides with either WT origin sequences or sequences corresponding to the M6, M7, M6, and M7 (M67) or M8 sequence substitutions and the indicated amount of UBP. Quantification of unbound DNA from the experiment shown in the upper panel of **d**, along with two further replicates. Bars indicate the average of the three replicates. Individual data points are shown for each replicate (black circles), and the error bar is the standard deviation from the average of the

three triplicates. Values for unbound DNA at 74 nM, 370 nM, and 740 nM UBP are shown in blue, orange, and green, respectively. **e** SDS-PAGE of the results of incubating 40 bp double-stranded annealed oligonucleotides with predicted Tm of 50.4 °C with increasing concentrations of UBP (0, 74, 370, or 740 nM) prior to incubation at 55, 60, or 65 °C and electrophoresis in the presence of SDS. Reactions that were boiled (Boil) or maintained at room temperature (RT) are included as controls for single-stranded (ss) or double-stranded (ds) DNA migration. **f** DNase I footprinting of increasing concentrations of UBP on 50 base-pair double-stranded DNA oligonucleotides containing the *ucm* site [top strand labeled (left panel) and bottom strand labeled (right panel), see sequence under gel images]. The size ladder on the right of the gel was generated by labeling a mix of synthetic oligonucleotides corresponding to sequential 10 nt 3' truncations of the substrates' labeled strands. The region of protection is summarized under the gel images. **g**. DNAse I footprinting on 181 bp *oriC1* sequences in the presence of Orc1–1 and UBP individually or in combination. A Maxam-Gilbert A + G sequencing ladder of the DNA substrate is shown in the left lane. Positions of ORB3, the *ucm* and ORB2 are indicated.

## Structure of UBP

To gain further insight into the function of the UBP protein, we determined its structure using X-ray crystallography. The structure of the protein, solved to 2.74 Å resolution, revealed a homodimer. The overall conformation is saddle-shaped with a basic, concave under-surface and more acidic on the outer, convex surface (Fig. 3a, b, Supplementary Figs. 2 and 3a, b). The concave surface is maximally 32 Å wide with the stirrups at the bottom of the saddle approximately 22 Å apart. Homodimerization is mediated by a reciprocal strand-swap of

the C-terminal beta strands of the protomers (Fig. 3a, c and Supplementary Fig. 2). This strand interacts with the neighbor, forming a 5-strand beta sheet. The interprotomer interface encompasses 1662.4 Å$^2$. We attempted to make a version of the protein lacking the C-terminal 30 amino acids, however it was chronically insoluble following cleavage from the MBP-tag. Examination of the structure reveals a potential π-stacking interaction between tyrosine 100 at the base of the C-terminal beta-strand of one protomer and phenylalanine 13 of its partner (Supplementary Fig. 3a). We mutated these residues to

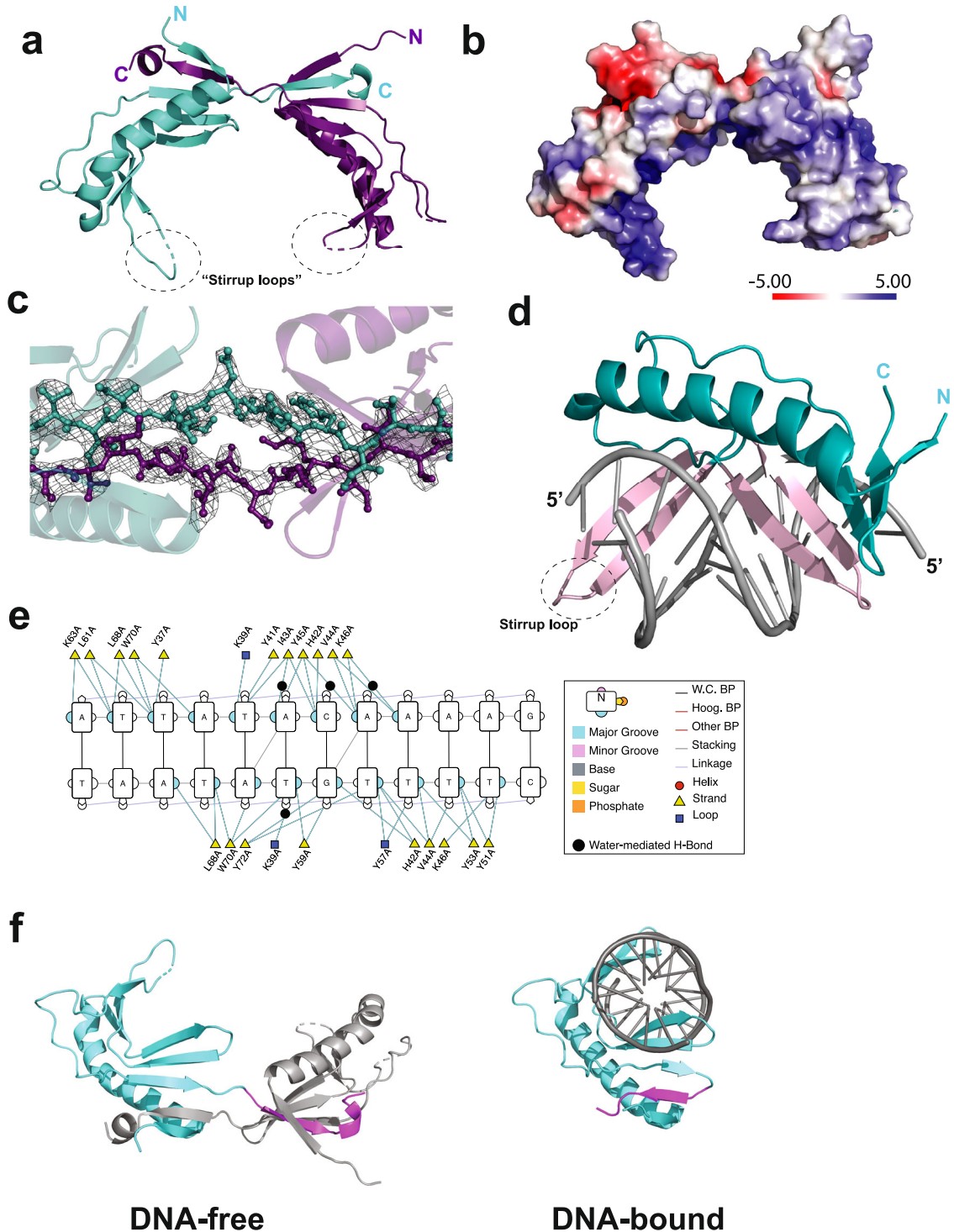

**Fig. 3 | Structural studies of free and DNA-bound UBP. a** X-ray crystal structure of the DNA-free UBP homodimer. One protomer in shown in purple, the other in teal. **b** Adaptive Poisson–Boltzmann solver electrostatic surface plot generated by Pymol v3.0.3 (Schrodinger, LLC) for the UBP homodimer. **c** Close-up of the C-terminal strand swap interface of the UBP homodimer, showing the electron density map as a mesh. **d** X-ray crystal structure of UBP bound to duplex oligonucleotides corresponding to the *ucm* sequence. The DNA-binding beta-hairpins are highlighted in lilac. **e** Contacts between UBP and DNA, figure generated using DNAproDB[50]. **f** Comparison from the same viewpoint of UBP in DNA-free and DNA-bound states. A single protomer is in teal with the C-terminal region of the protein that undergoes strand-swap in the absence of DNA highlighted in purple.

alanine and in both instances observed an altered profile on gel filtration chromatography in agreement with destabilization of the dimer and an elevated population of monomer species (Supplementary Fig. 3d). Interestingly, both UBP Y100A and UBP F13A displayed similar DNA binding properties to the wild-type protein. However, substitution by alanine of three lysine residues (K63, 67, and 69) in the stirrup loops of the protein has no effect on the dimeric status of the protein but abrogates DNA binding (Supplementary Fig. 3b–e). Mutation of another conserved lysine, K77, on the outer surface of the saddle had no effect on either multimerization or DNA binding activity

(Supplementary Fig. 3d, e). We additionally solved a second structure of UBP from crystals that had a distinct space group (P21). Interestingly, these crystals, which were twinned, had four dimers of UBP in the asymmetric unit. Two dimers were essentially identical to the initial structure (C2221); the other two showed the same dimerization interface, but the subunits had undergone a ~10° rotation with respect to one another (Supplementary Fig. 3f, g).

Next, we solved the structure of UBP bound to DNA containing a *ucm* site to 1.8 Å resolution. We attempted crystallization with a range of sizes of DNA molecule; the only construct that yielded crystal formation was a 12 nt duplex DNA (Supplementary Fig. 4a, d). Significantly, and in contrast to the homodimeric free protein, UBP binds DNA as a monomer (Fig. 3d). The interaction with DNA is principally via the major groove (Fig. 3d) and is mediated by two sets of beta-hairpins within the structure, one corresponding to the stirrup loops in the free protein and the other being a part of the 5-beta-strand sheet to which the C-terminal beta strand contributes. Of the three lysine residues in the stirrup loop, K63 makes base-specific contacts in the protein-DNA co-crystal structure (Fig. 3e). The DNA structure remains close to standard B-form but with a widening of the minor groove to 8 Å at the center of the molecule. Notably, the beta-strand that undergoes the intermonomer strand swap in the DNA-free dimer is now observed to fold back to form an analogous interaction in *cis* (Fig. 3f). Furthermore, the interprotomer Y100−F13 π-stacking interaction switches from an offset sandwich orientation in the dimer to a T-shaped interaction in the snapped-back *cis* configuration of the C-terminal beta strand in the DNA-bound monomer (Supplementary Fig. 4c).

## UBP interacts with the N-terminal domains of the MCM replicative helicase

Our data indicate that UBP interacts specifically with the *ucm* motif within the DUE, midway between the ORB2 and ORB3 Orc1-1 binding sites. Our previous data indicate that these sites, when bound by Orc1-1•ATP, direct the recruitment of MCM to the DUE region[12]. Thus, we speculated that UBP might interact with MCM. We first performed yeast 2-hybrid assays with UBP, ORC1-1 and MCM. The data indicated an interaction between UBP and MCM (Fig. 4a). We next prepared 2-hybrid constructs with the N-terminal (residues 1–266), AAA+ (residues 267–601) and C-terminal winged helix (wH, residues 602–686) domains of MCM (Fig. 4b, c). We observed interactions between UBP and the N-terminal domains of MCM. Next, we split MCM's NTD into its constituent A and B/C domains (Fig. 4b) and could detect interactions between UBP and both of these constructs (Fig. 4d). We then performed GST-pulldowns in which MCM's NTDs or constituent subdomains were fused to Glutathione-S-transferase, confirming interactions of UBP with both A and B/C domains of MCM. Initial gels detected proteins with Coomassie Brilliant Blue and provided hints that UBP interacted; however, due to a co-migrating fragment of GST, we additionally performed western blotting with anti-UBP antisera on the pulldowns, confirming the interaction between MCM's NTDs and UBP (Fig. 4e).

While the above assays indicate that UBP and MCM can interact in solution, we additionally wanted to test whether they could interact with DNA. To this end, we performed electrophoretic mobility shift assays (EMSAs) with either MCM or the MCM NTDs following preincubation of the DNA in the presence or absence of UBP. As we have shown previously, MCM and MCM NTDs form complexes with DNA[24]. Importantly, we observed preferential shifting of UBP−DNA complexes by MCM and MCM NTDs over naked DNA, indicative of interactions between MCM's NTDs and UBP on the DNA molecule (Fig. 4f, g).

## UBP in vivo

We raised antisera against the UBP protein and performed western blotting using whole cell extracts and known amounts of recombinant proteins (Fig. 5a). Analysis of these data gave an abundance of

approximately 5400 copies of UBP per cell, which, based on an average cell diameter of 1 μM and assuming perfectly spherical cells, gives a cellular concentration of approximately 13 μM. Western blotting of extracts prepared from various timepoints during the growth of a culture reveals UBP to be present in exponentially growing cells but to decrease markedly in abundance following entry into stationary phase (Fig. 5b).

We performed chromatin immunoprecipitation assays coupled with quantitative PCR on extracts from crosslinked and sonicated *S. islandicus* REY15A wild-type and M7-mutant cells with antisera generated against UBP and MCM. qPCR was performed with primer pairs specific for *oriC1* and *oriC3*, the data values shown in Fig. 5c are the ratio of DNA recovered at *oriC1* divided by that at *oriC3*. In agreement with the biochemistry described above, the M7 mutants show highly reduced origin occupancy by UBP compared to the recovery of origin DNA from wild-type cells. Intriguingly, however, ChIP with anti-MCM antisera showed 3.5-fold elevated recovery of origin DNA in the M7 mutant compared with wild-type (Fig. 5c). This was surprising in light of our above observations indicating an interaction between MCM and UBP. However, prolonged origin-occupancy by MCM would result in an elevated recovery of DNA in the ChIP assays. Thus, these data suggest that UBP's role at the replication origin is not in the recruitment of MCM but, rather, in facilitating MCM's exit from the origin. We additionally profiled Orc1–1 occupancy in the WT, M7, and M12 (ORB-mutated strain). Importantly, the M7 mutation did not impact on Orc1–1's ability to interact with the origin. In contrast, a reduction in Orc1–1 ChIP signal was observed in the M12 strain in which the Orc1–1 binding site, ORB2, has been mutated.

## UBP binds to many sites in the chromosome

Next, we sought to determine the genome-wide localization of UBP binding sites on the *S. islandicus* chromosome by performing ChIP-Seq experiments with anti-UBP antisera. Remarkably, MACS peak caller identified 329 significant peaks, corresponding to genomic loci occupied by UBP (Fig. 5d, upper panel). Importantly, a parallel analysis using the M7 strain, with a mutated *ucm*, showed a high correspondence of all peaks, with the exception of the *oriC1*-associated signal ($R = 0.976$, Fig. 5d, lower panel and 5e, Supplementary Fig. 5a). We have previously characterized architectural features on the *S. islandicus* chromosome in the form of A and B compartments, local domains termed CIDs and gene loop structures[17,25]. UBP was significantly elevated in the A compartment (Supplementary Fig. 5b, c). We could not observe any significant association between UBP binding and the location of CID boundaries or gene loop anchors. UBP binding sites were enriched in intergenic regions (Fig. 5f), however, we could not observe any correlation between UBP peak level at its binding loci and the transcriptional status of adjacent genes beyond the general A-compartment enrichment. Analysis of the binding loci with the MEME suite[26] revealed a consensus sequence with perfect correspondence to the *ucm* site identified at the three replication origins (Fig. 5g).

The abundance of the UBP protein and the large number of binding sites to genes of disparate families suggest that the protein may be functioning as a nucleoid-associated protein. The gene for UBP is essential, and our attempts to knock down expression using the endogenous RNA-targeting CRISPR system of *Sulfolobus* were unsuccessful. However, it was possible to over-express the gene from a plasmid in *S. islandicus* (Supplementary Fig. 6a). Analysis of culture growth revealed only modest differences from the empty vector control in early or mid-exponential growth (Fig. 6a). However, while the control strain ended exponential growth at 45 h, the increase in optical density continued for an additional 10 h in the presence of elevated UBP, before peaking at 55 h. Flow cytometry revealed a slight enhancement of G1 (1 C) cells in the UBP over-expressing cultures at 30 h of UBP induction compared to the empty-vector containing

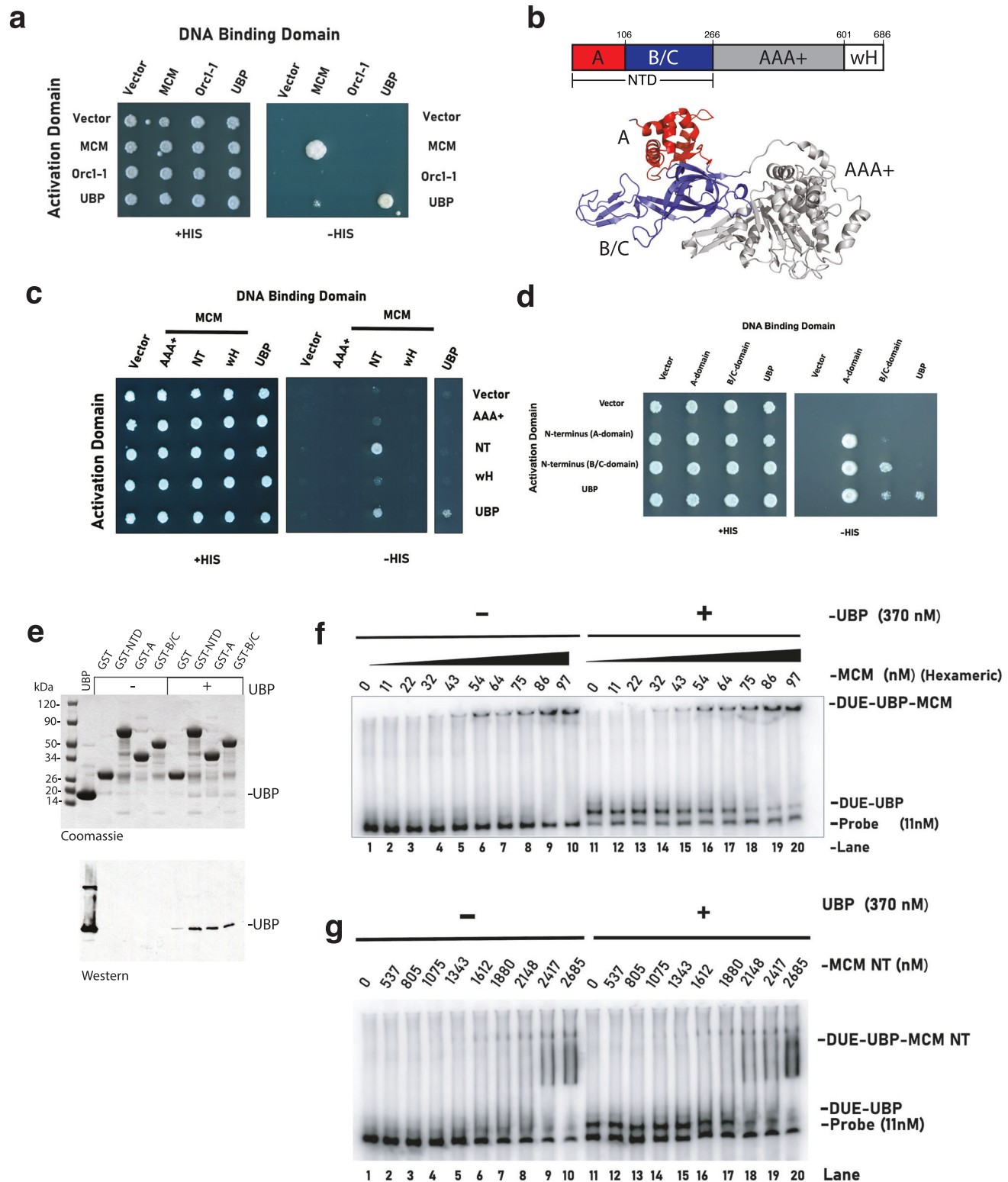

cultures (Fig. 6b and Supplementary Fig. 6b), indicating ongoing cell division in the UBP over-expressing cells at this time point. However, there were no discernible alterations to cell cycle distribution between the two populations at the 45- and 55-h time points, suggesting the increase in optical density upon UBP overexpression at these time points is not due to a prolonged period of cell division, but rather due to enhanced cell growth. Analysis of the forward scatter profile from the flow cytometry data reveals a consistent increase in scatter by cells containing the UBP over-expression construct, indicating that these

cells are reproducibly larger than those containing the empty vector. We performed transcriptomic profiling of cultures containing either an empty vector or the UBP over-expression plasmid. Duplicate cultures were grown in the presence of the inducer, arabinose, and RNA was extracted at 30, 45, and 55 h after induction; for the empty vector-containing cell these time points were observed to correspond to mid-exponential growth, end of exponential growth/onset of stationary phase and late stationary phase (Fig. 6a, c–e). RNA-Seq results on the empty vector-containing strain reveal the anticipated extensive

**Fig. 4 | UBP interacts with the N-terminal domains of MCM. a** Yeast two-hybrid (Y2H) analysis of interactions between UBP, Orc1–1 and MCM, fused to either DNA binding domain or transcriptional activation domain. Growth on His plates is indicative of interaction. **b** Diagram of the domain organization of MCM (upper panel) and the X-ray crystal structure with the A, B/C and AAA+ domains colored as in the above diagram. The C-terminal wH domain is not present in the crystal structure[40]. The figure was generated from PDB file 3F97 using Pymol v3.0.3 (Schrodinger, LLC). **c** Y2H assays between UBP and the constituent N-terminal (NT, residues 1–266), AAA+ (residues 267–601) and C-terminal winged-helix (wH, residues 602–686). **d** Y2H assays with sub-domains of MCM's N-terminal domains–A (1–106) and B/C (107–266). **e** GST pull-down assays with GST alone, GST fused to MCM 1–266 (GST-MCM-NT), MCM 1–106 (GST-A) or MCM 107-266 (GST-B/C). The

upper panel shows reactions run on a gel and stained with Coomassie, the lower panel is the result of western blotting of these samples, detected with anti-UBP antisera. **f** EMSAs with 140 bp *ucm*-containing oligonucleotides derived from oriC1. Reactions were incubated in the presence or absence of 370 nM UBP before the addition of increasing concentrations of MCM prior to electrophoresis on a native gel. The positions of free DNA, UBP-DNA complex and DNA-MCM or DNA-UBP-MCM complex is indicated. **g** EMSAs with 140 bp *ucm*-containing oligonucleotides derived from *oriC1*. Reactions were incubated in the presence or absence of 370 nM UBP before the addition of increasing concentrations of MCM's N-terminal domain (aa 1–266, MCM–NT) prior to electrophoresis on a native gel. The positions of free DNA, UBP–DNA complex and DNA–MCM–NT, or DNA–UBP–MCM–NT complex, are indicated.

remodeling of the transcriptome over this time course[17], in agreement with the profound physiological changes experienced as cells enter into, and persist in, stationary phase. In contrast, over-expression of UBP results in the maintenance of a gene expression profile more closely related to that of the exponential phase, accounting for the extended period of growth seen in the over-expression strain. DESeq2 analysis identifies 739 genes as being significantly ($p < 0.05$) altered in the data set as a whole (Fig. 6e). Interestingly, comparison of vector-only and UBP-per-expression strains at given time points (Supplementary Fig. 7, Supplementary Data 1) reveal 168 differentially expressed genes, of which 66 lie within a kilobase of a UBP binding site in wild-type, exponentially growing cells. At 45 h, 168 genes are distinctly expressed with 66 within a kilobase of a UBP binding site. Finally, at 55 h, 360 genes are differentially expressed and of those, 135 lie within one kilobase of a UBP binding site. Taken together, our data therefore implicate UBP as a key modulator of growth-phase dependent gene expression in *Sulfolobus*.

Finally, we addressed the role that UBP plays at the replication origin by performing in vitro loading assays with purified recombinant proteins (Fig. 7a). As we have described previously, the E147A Walker B mutant (ATP-binding proficient, hydrolysis deficient) of Orc1–1 recruits MCM to replication origins[12]. While the majority of MCM recruited by Orc1–1 E147A is in a salt-sensitive state, a sub-population forms a high-salt stable complex with DNA. Although we have not yet established the structures of these differentially salt-sensitive forms of origin-associated MCM, based on our previous work, we believe that the salt sensitive form of MCM in these assays is protein that has been recruited to the origin via interactions with Orc1–1 but not yet stably associated with DNA (Stage II in Fig. 7b). The salt-stable form is proposed to be MCM which now has DNA present within its central channel (e.g., Stage III in Fig. 7b). In agreement with the interaction studies described above, UBP is able to form a complex with MCM on DNA, however, the level of MCM recruited is lower than seen with Orc1–1. Notably, essentially all the MCM recruited by UBP is in a high-salt-stable form. The combination of Orc1–1 and UBP results in similar levels of high-salt stable MCM to those observed with UBP and MCM alone.

## Discussion

Our data indicate complex and multiple roles for UBP in *Sulfolobus* chromosome biology. The over 300 binding sites identified by ChIP-Seq and the effects of modulating UBP expression on the transcriptome and cell growth indicate a role as a general nucleoid-associated protein (NAP). To date, the other *Sulfolobus* NAPs that have been characterized, such as Alba, Cren7, and the Sul7 family, appear to be sequence non-specific in their DNA binding[27]. In contrast, UBP has a well-defined consensus sequence. In this regard, conceptual parallels might be drawn with the *E. coli* NAPs IHF and Fis, both of which play roles at the *E. coli* DNA replication origin[7]. Like UBP, IHF and Fis are highly abundant proteins during logarithmic growth (5000–10,000 and 50,000–100,000 copies per cell respectively), which bind to

defined consensus sequences and modulate the expression of hundreds of genes (for review see ref. 7). While IHF levels are maintained in stationary phase, Fis, like UBP, drops to undetectable levels in stationary phase. Fis is also important for the regulation of over 200 genes in the *E. coli* chromosome, and is of particular importance in growth-phase dependent gene regulation[28]. However, while there are clear functional parallels between UBP and *E. coli* NAPs, it must be stressed that there are no structural similarities between IHF or Fis and UBP. The bacterial NAPs dramatically alter the structure of DNA upon binding (IHF induces a bend of 160°, Fis approximately 60°). The profound bend induced by IHF has been proposed to be key to establishing a functional architecture at the *E. coli* oriC[7]. In contrast to IHF and Fis, although a modest widening of the minor groove is seen, UBP does not introduce any discernible bending of DNA in our protein-DNA co-crystal structure. One fundamental difference between bacterial and archaeal/eukaryotic DNA replication systems is that in *E. coli*, the initiator protein, DnaA, in concert with IHF, is capable of melting the replication origin, leading to the assembly of the replicative helicase on single-stranded DNA[7]. In contrast, all data to date point to the MCM replicative helicase being loaded onto double-stranded DNA in archaea and eukaryotes[29]. In eukaryotes, two hexamers of MCM(2–7) are loaded onto replication origins and form a stable double-hexamer via interaction between the juxtaposed N-terminal domains of the hexamers[30–33]. Structural studies with budding yeast proteins have revealed that DNA melting occurs within the inner chambers of the constituent hexamers of the double-hexamer. Ultimately, the two hexamers must split apart in order to effect their departure from the replication origin and allow establishment of the two replication forks necessary for bidirectional replication. Structural and biochemical studies have demonstrated that a topological consequence of the localized unwinding of DNA within each MCM(2–7) hexamer is that compensatory over-winding of DNA accumulates between the two hexamers[34–36]. The dissolution of the double-hexamers to individual hexamers is thus of profound importance in that it permits the two hexamers to rotate relative to one another, dissipating this localized over-winding. The initiation factor MCM10 has been demonstrated to play a role in this separation of the two hexamers[34,35]. Intriguingly, however, MCM10 appears to be a eukaryotic innovation; no archaeal orthologs of this factor have been identified. In archaea, while structural studies of *Methanothermobacter thermautotrophicus* MCM support formation of a double-hexamer in the absence of DNA[37,38], the *Sulfolobus* MCM species appear to exist primarily as a single hexamer[24,39–41]. Whether archaeal MCMs form double hexamers at origins and, if they do, how the topological issues discussed above would be dealt with, remain unresolved issues.

Our previous data support a model in which ORB2 and ORB3 direct the recruitment of open hexamers of MCM to the intervening DUE region[42]. The C-terminal wH domain of MCM interacts with the ORB-bound Orc1–1, thus setting up a geometry where N-terminal tiers of the MCM hexamers face one another. However, our current data indicate that the *UCM*, located midway between the ORB elements, is

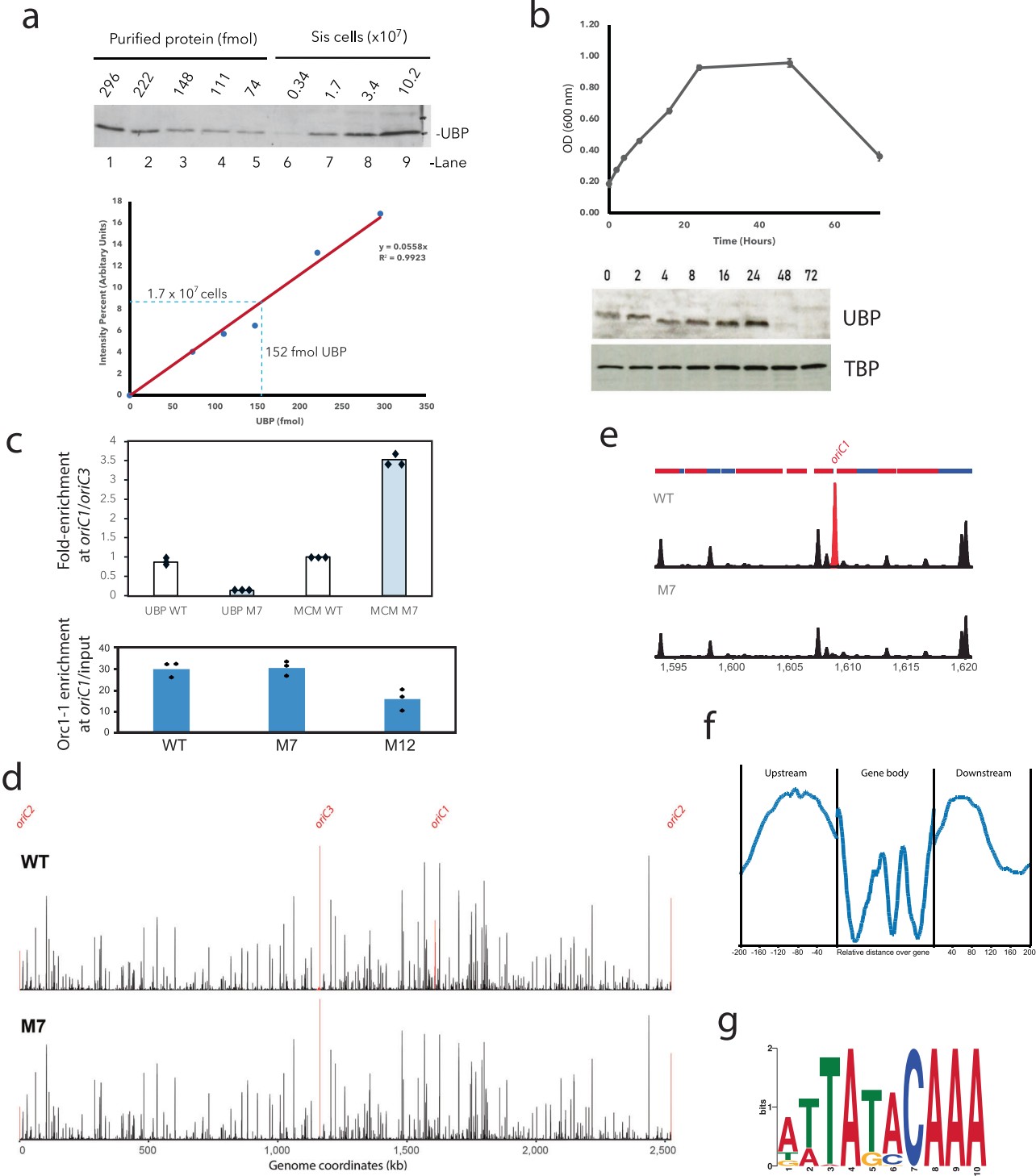

**Fig. 5 | UBP expression, binding profile and the role of the *ucm* in vivo.**
**a** Western blot (upper panel) of a titration series of purified recombinant UBP (lanes 1–5) and whole cell extract from known numbers of cells (lanes 6–9). The lower panel indicates a graph of signal intensity versus UBP quantity with a linear curve fit applied. The blue line indicates a signal intensity for UBP in extract from $1.7 \times 10^7$ cells, which is clearly in the linear range, and an extrapolation to the x-axis to reveal the corresponding amount of UBP. **b** Growth, stationary and death phases of a *S. islandicus* culture monitored by OD6000 nm. Western blotting of whole cell extracts at the indicated timepoints with antisera to UBP and, as a loading control, the general transcription factor, TBP. **c** Upper panel: chromatin precipitation experiments with anti-sera against UBP and MCM in the wild-type and M7-mutation-containing strain. qPCR was performed with primer pairs specific for *oriC1* and *oriC3*, the bar graphs indicate the mean of three independent experiments with the values of recovery of input-normalized *oriC1* DNA divided by that of

input-normalized *oriC3* DNA. Individual data points are shown as diamonds. ChIP results from the wild-type strain are shown in white, from M7 in pale blue. The lower panel shows ChIP experiments using anti-Orc1–1 antisera in the WT, M7 and M12 strains, Values are normalized to input DNA, as above; the bars represent the mean of three independent experiments, with individual data points shown as diamonds. **d** A genome-wide profile of UBP binding assayed by ChIP-Seq for wild-type and M7 mutant strains, the positions of the three replication origins are indicated in red. **e** Enlarged view of the *oriC1*-containing region of the chromosome for the data shown in panel (**d**). **f** Aggregated analysis of the relative abundance of UBP ChIP signal in regions up to 200 bp upstream of open-reading frames (ORFs), across the ORFs themselves and in the downstream 200 bp. **g** A consensus sequence was identified using the MEME suite[26] using a sequence library of the top 100 ranked UBP sites identified by the MACs peak caller.

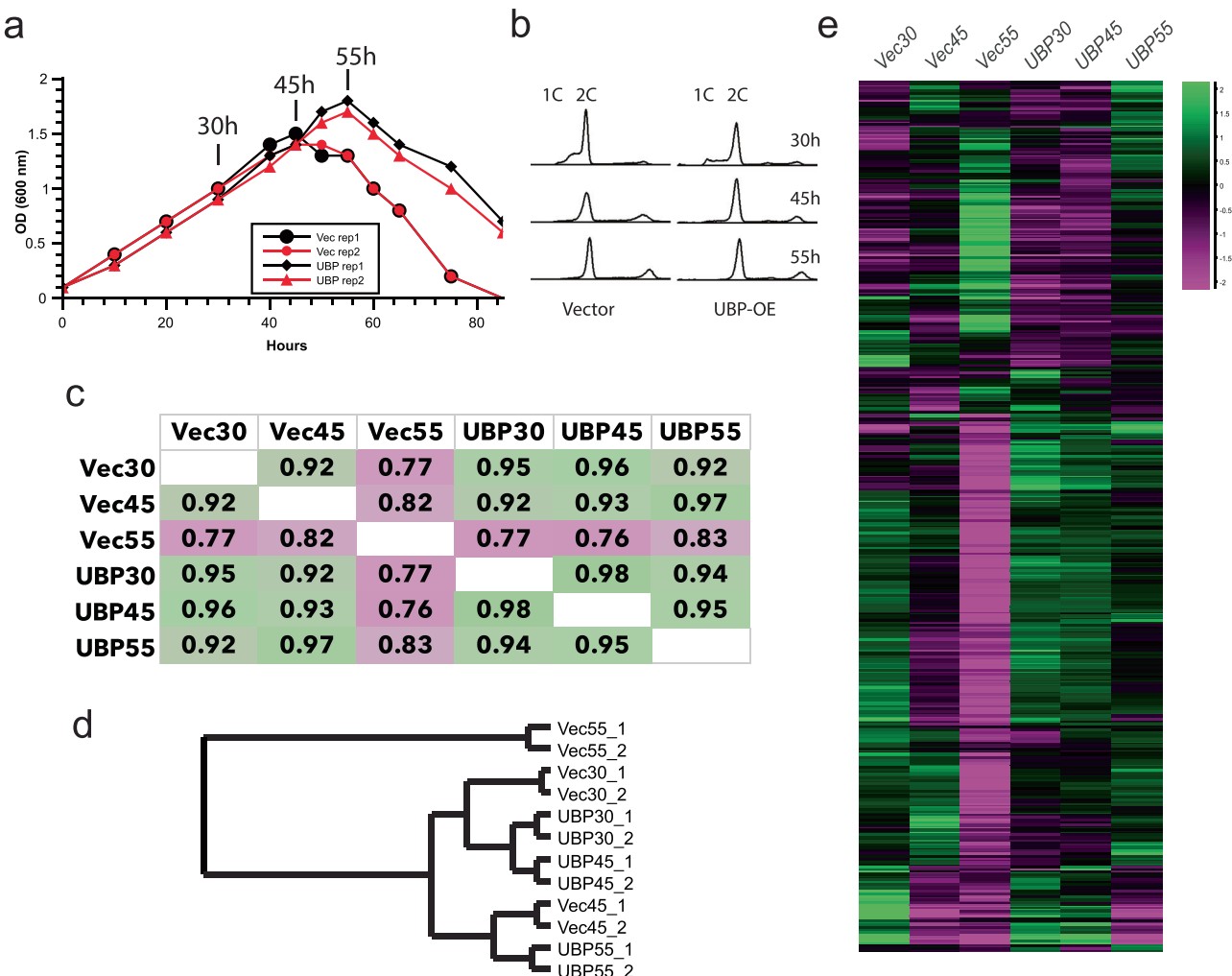

**Fig. 6 | Phenotypic consequences of UBP overexpression. a** Growth curves of *Sulfolobus islandicus* cells containing either a UBP-over-expression plasmid (diamond and triangle) or the empty vector control (circles) grown in the presence of arabinose. Duplicate cultures are indicated by red and black and the absorbance at 600 nm is plotted. Time points sampled for the subsequent experiments are highlighted (30, 45 and 55 h). **b** Flow cytometry profiles of cells containing either a UBP-over-expression plasmid (UBP-OE) or the empty vector control (vector) grown in the presence of arabinose/see also Supplementary Fig. 6. **c** Pearson correlation matrix of gene expression profiles of the stains (empty vector−"Vec" or UBP-over-expressing "UBP") at the indicated time points (30, 45, and 55 h after arabinose addition). Figure generated using SeqMonk (Babraham Institute, UK). **d** Neighbor joining tree derived from the Pearson correlation distance matrix in panel (**c**). Figure generated using SeqMonk (Babraham Institute, UK). **e** Heat map of expression profiles of the 793 coding sequences that were identified by DESeq2 as altering significantly (*p* < 0.05) in the dataset. Plot was generated using the hierarchical profile option in SeqMonk (Babraham Institute, UK). DESeq2 was performed with a two-sided test with multiple testing correction applied.

bound by the UBP protein. Furthermore, we have found that UBP is capable of interacting with the N-terminal domains of MCM, both in solution and, importantly, when bound to DNA. Thus, we propose that UBP serves as a bridge or spacer between the two single hexamers of MCM on the replication origin. Our initial determination of the homodimeric structure of UBP in the absence of DNA suggested the establishment of a symmetric structure of UBP at the origin. However, our demonstration that UBP binds the *ucm* as a monomer suggests this is not the case. It is currently unknown whether a single UBP contacts both MCM hexamers or whether UBP imparts a functional asymmetry to the MCM activation process. Assuming the MCM−UBP−MCM geometry is established, we speculate that initial DNA unwinding occurs within one or both of the MCM hexamers. By analogy with the budding yeast system[34,35], the topologically inevitable compensatory overwinding may be directed between the individual hexamers, more specifically to the *ucm* sequences. We suggest that this alteration of local double-helical geometry at the *ucm* would lead to displacement of UBP from the DNA, thus the MCM hexamers would now be free to

rotate with respect to each other. This speculative model is compatible with our observations that while the *ucm* is essential for origin function, UBP plays a minor role in MCM recruitment in in vitro loading assays and the accompanying apparently paradoxical observation that mutation of the *ucm* leads to elevated recovery of *oriC1* DNA in MCM ChIP assays. However, it is also possible that UBP plays other or additional roles in the post-recruitment activation and origin-exit of MCM. Finally, it is intriguing that the phyletic distribution of the *ubp* gene suggests that, in evolutionary terms, it is a relatively recent acquisition in the *Desulfurococcales* and *Sulfolobales*. It will be of considerable interest to determine if other, more divergent archaeal lineages have analogous proteins functioning at their replication origins.

## Methods

### Strains, media and cell growth conditions

The wild-type strain (*Sulfolobus islandicus* E223S−*S. islandicus* REY15A Δ*pyrEF*Δ*lacS*) and *ucm* mutant strains (UCM-m1 to m13) were grown at

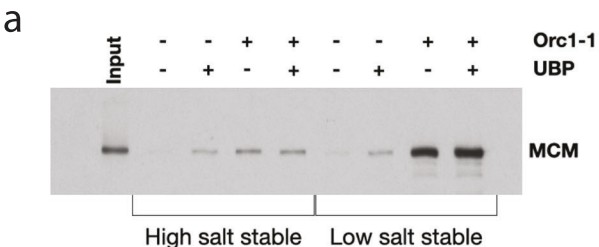

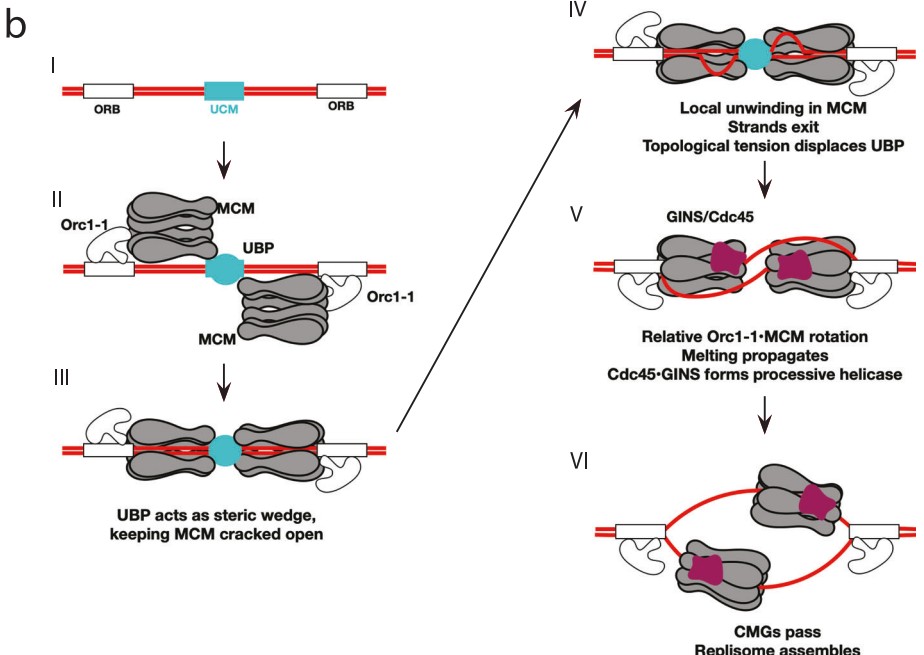

**Fig. 7 | Model for UBP function at *oriC1*. a** In vitro loading assays using purified recombinant Orc1–1 (E147A), MCM and UBP as indicated. **b** Speculative model for the role of UBP and the *ucm* at *oriC1*.

78 °C in SCV media (Basal media supplemented with 0.2% sucrose, 0.2% Casamino acids and 1% vitamin solution) supplemented with uracil (20 μg/ml) as described previously[43].

### Construction of *S. islandicus oriC1* UCM mutant strains

A genome editing method harnessing the endogenous type IA CRISPR system was utilized to construct *oriC1* UCM mutant strains[44]. The 130 bp UCM-containing *oriC1* was divided into 13 regions (namely M1, M2 through to M13), which were mutated individually by introducing base transversions. Briefly, a type IA protospacer adjacent motif (CCN or TCN) and its downstream 40 nt DNA sequence were chosen as the target site for each region of the UCM. Spacer DNA fragments were generated by annealing two complementary oligos (Table S1, for example, M1spf and M1spr) and were inserted into the artificial mini-CRISPR array of pGE1 vector at SapI site[45], giving the interference plasmids, pAC-M1 through to pAC-M14. Then, donor DNAs used to introduce the mutated alleles were prepared individually using SOE PCR with primer pairs indicated in Table S1 (For example, ucmDonor-f/M1SOEr and M1SOEf/ucmDonor-r) and cloned into the corresponding interference plasmid at the SphI and XhoI, giving genome editing plasmid pGE-UCM-M1 to pGE-UCM-M13. These plasmids were then transformed into the *S. islandicus* REY15A (Δ*pyrEF*Δ*lacS*) competent cells individually by electroporation. Transformants containing mutated target alleles were examined by Colony PCR and DNA sequencing of the resulting PCR product. Genome editing plasmids in the positive

transformants were cured by 5-FOA counterselection, yielding UCM mutant strains for subsequent analyses.

### Flow cytometry

*Sulfolobus islandicus* cells were fixed with 72% ethanol, stained with SYTOX Green (Invitrogen) and DNA contents were run on an LSRII flow cytometer in the Flow Cytometry Facility at Indiana University and analyzed with FCS Express 7 software (De Novo Software).

### Chromatin immunoprecipitation

Chromatin immunoprecipitation (ChIP) was performed as described[17]. Early logarithmic cultures of *S. islandicus* were crosslinked with 1% formaldehyde for 20 minutes. After quenching with 125 mM glycine, the cells were pelleted and washed with 1× PBS. The pellets were resuspended in TBS-TT (20 mM Tris, 150 mM NaCl, 0.1% Tween-20, 0.1% Triton X-100, pH 7.5) and sonicated using a Diagenode Bioruptor to generate DNA fragments ranging from 200 to 1000 bp. The extract was then clarified by centrifugation. 10 μg of extract (based on protein concentration) were used in each 100 μl ChIP reaction. Samples were rotated for 2 h with 3 μl of antiserum at 4 °C. In total, 25 μl of a 50% slurry of protein A sepharose were then added and the samples were rotated for another hour at 4 °C. Each ChIP reaction was then washed five times at room temperature with TBS-TT, once with TBS-TT containing 500 mM NaCl, and once with TBS-TT containing 0.5% Tween-20 and 0.5% Triton X-100. Protein–DNA complexes were eluted from

the protein A sepharose in 20 mM Tris, 10 mM EDTA, 0.5% SDS, pH 7.8 at 65 °C for 30 min. Crosslinking was reversed and protein was digested by incubating the samples with 10 ng/μl Proteinase K for 6 hours at 65 °C followed by 10 h at 37 °C. The samples were extracted with phenol/chloroform/isoamyl alcohol first, then chloroform alone and the DNA was precipitated in 100% ethanol containing 20 μg of glycogen. After washing with 70% ethanol and air drying, the DNA was resuspended in 50 μl TE buffer.

## Purification of endogenous UBP from *Sulfolobus acidocaldarius*

*Sulfolobus acidocaldarius* cells (5 × 1 l) were grown at 75 °C. Cells were collected by centrifugation at 3200×*g* for 20 min. Cell pellets were resuspended in 150 ml of lysis Buffer AC1 (50 mM HEPES pH 7.5, 100 mM NaCl 10 mM DTT) supplemented with 0.1% Tween 20 and 0.1% Triton X-100 and mixed using a rotating mixer. Throughout the purification protocol, UBP-containing fractions were selected based on the ability to bind the UCM30 sequence in gel shift assays. After centrifugation at 28,000*g* for 30 min, the supernatant was collected and loaded onto a 5 ml HighTrap Q Fast Flow column (Cytiva) equilibrated with Buffer AC1. The flow-through was collected and loaded onto four 1-ml Hitrap Heparin HP columns (Cytiva), assembled in series. To maximize resolution during elution, the four heparin columns were detached, and four individual elution steps were performed over 10 column volumes using a 150 mM-1M NaCl gradient. Five 1 ml fractions eluting at ~300 mM NaCl were pooled and diluted 2-fold, resulting in final solution composition (25 mM HEPES pH 7.5, 150 mM NaCl, 100 μM ZnCl, 10 mM MgCl, 10 mM DTT, 5% glycerol, 0.1% Tween 20, 200 μg/ml salmon sperm DNA). For preparation of the DNA affinity matrix, 1 mg of streptavidin-coated paramagnetic beads (Dynabeads M-280) were resuspended in 2× B&W Buffer (10 mMTris-HCl pH 7.5, 2 M NaCl and 1 mM EDTA) to a final concentration of 5 μg/μl, after washing them once with 500 μl of 1× B&W Buffer. For immobilization of UCM or control (scrambled) DNA, 800 μl of 1× B&W Buffer and 10 μg of biotinylated DNA were added and incubated for 30 °C at room temperature. Double stranded UCM-containing oligonucelotides were generated by annealing UCM30TOP (5'GCTTGGTTTA**ATTATACAAAC**TCAATATTT) and UCM30BOTbiotin 5'[Biotin]AAAATATTGA**GTTTGTATAAT**TAAACCAAGC and the control DNA made by annealing CTRLTOP GCTTGGTTTA**CGGCGCACCCA**TCAATATTT and CTRLBOT [Biotin]AAATATTGA**TGGGTGCGCCG**TAAACCAAGC. The wild-type or mutated *ucm* sequences are highlighted in bold and underlined.

After magnetic separation, the affinity beads were washed three times in 1 ml 1× B&W Buffer and resuspended in 150 μl of 25 mM HEPES pH 7.5, 150 mM NaCl, 100 μM ZnCl, 10 mM MgCl, 10 mM DTT, 5% glycerol, 0.1% Tween 20. DNA-tethered magnetic beads were incubated at 48 °C for 10 min in three subsequent binding and wash steps. The bead-bound proteins were analyzed using SDS-PAGE followed by staining using the Pierce Silver Stain kit for Mass Spectrometry kit (Thermo Fisher). Only two bands were visible after development (one unique to the UCM DNA bait, the other also present in the scrambled control DNA bait). Bands were excised and sent for elution from the gel and submitted to the Sir William Dunn School Proteomics Facility, Oxford University for trypsinization and protein identity determination through LC–MS/MS analysis.

## ChIP-Seq

ChIP reactions were performed in duplicate and pooled. In total, 50 μl of ChIP reactions and 100 ng of input DNA were used to construct DNA libraries using Illumina TruSeq ChIP Library Preparation Kits. The manufacturer's instructions were followed, except DNA size selection was performed using a 0.6× ratio of AMPure XP beads rather than purification by agarose gel. DNA quality, size and quantity were verified using a High Sensitivity DNA chip on an Agilent Technologies 2100 Bioanalyzer and with a Qubit dsDNA High Sensitivity Assay kit on a Qubit 2.0 Fluorometer (Invitrogen). Libraries were sequenced using

the Illumina NextSeq platform in the Center for Genomics and Bioinformatics at Indiana University. Paired-end sequence reads were mapped onto the *S. islandicus* genome using Geneious software (http://www.geneious.com). The resulting data were exported as.bam files and subsequently analyzed using SeqMonk (http://www.bioinformatics.babraham.ac.uk/projects/seqmonk/). Read counts were normalized to input DNA and quantified in 20 bp windows.

## Quantitative PCR

SYBR detection assays were performed in triplicate using 1 μl of ChIP DNA as template, forward and reverse primers at 125 nM final concentration, and Luna Universal qPCR Master Mix (New England Biolabs) according to the manufacturer's instructions. Standard curves were prepared in duplicate using ChIP input DNA as template. qPCR reactions and melting curves were performed and analyzed using a Mastercycler ep realplex[2] machine and included analysis software (Eppendorf).

## Marker frequency analysis (MFA-Seq)

DNA was purified from exponentially growing ($OD_{600} = 0.2$) *S. islandicus* E233S and *oriC1* mutant strains and E233S in stationary phase. Libraries were prepared for whole-genome sequencing using Nextera DNA Flex kits (Illumina) and sequencing was performed on the Illumina NextSeq platform in the Center for Genomics and Bioinformatics at Indiana University. At least 10 million reads were obtained for all samples and read counts in 1 kb bins from exponentially growing samples were normalized to the corresponding read count bins from the *S. islandicus* E233S stationary phase sample.

## UBP cloning, expression, mutagenesis and protein purification

The *Sulfolobus islandicus* REY15A, *ubp* gene (SiRe_1573), a 336 bp DNA fragment codes for 111 amino acids, was amplified by primers (P1 and P2) and cloned into the *NdeI* and *SalI* sites of pMAL-c5X plasmid vector. BL21DE3 Rosetta *E. coli* cells transformed with this construct were grown at 37 °C in 1 L culture, and the induction of (55.8 kDa) MBP–UBP fusion protein expression was initiated at $OD_{600nm} = 0.8–0.9$, with 1 mM IPTG, for 5 h.

Cells were harvested by centrifugation, yielding 5 g of cell pellet from 2 L of culture. This was resuspended and lysed for 30 minutes in the 40 ml lysis buffer containing 5 ml TBS buffer (10 mM Tris pH 8.0, 150 mM NaCl), 35 ml BugBuster protein extraction reagent (Millipore), 50-μl Benzonase (Millipore) and Complete mini, EDTA-free protease inhibitor cocktail (Roche). Following the centrifugation of cell lysate (4000×*g* for 30 min at 4 °C), the supernatant was incubated with 5 ml Amylose-resin beads (New England Biolabs) at 4 °C for 60 min. The column was first washed with high salt buffer (10 mM Tris pH 8.0, 500 mM NaCl) and followed by TBS buffer, and later the bound MBP–UBP protein was eluted by 30 ml of elution buffer containing 10 mM maltose in TBS. The eluted protein solution was concentrated using 30 kDa cutoff concentrator and then injected into Hi Load 26/600 Superdex 75 gel filtration chromatography with TBS buffer. The 30 ml pooled peak fractions containing MBP–UBP fusion were supplemented with 2 mM $CaCl_2$ and treated with 20 μl (1 mg/mL) Factor-Xa protease (New England Biolabs) for 8–12 h at room temperature. The N-terminus of the UBP gains three additional amino acids from the vector after its separation from the cleaved MBP tag. The cleaved products: MBP-tag (42 kDa) and UBP (13.9 kDa), were separated over size-exclusion chromatography, Hi Load 16/600 Superdex 75 column (GE Healthcare) in TBS buffer. The UBP peak fractions were combined and concentrated in a 5-kDa cut-off concentrator.

UBP's indicated site-directed mutants (F13A, Y100A, K77A, K63,67,69A) were generated on the pMAL-c5X-UBP expression vector plasmid by primer extension mutagenesis method using respective mutation primers (P31, P32, P33, and P34) (Table S1) The expression

**Table 1 | Data-collection and refinement statistics**

| | UBP WT Crystal 1 | UBP WT Crystal 2* | UBP–DNA |
|---|---|---|---|
| *Data collection* | | | |
| Wavelength (Å) | 1.24984 | 1.24984 | 0.91165 |
| Space group | C2221 | P21 | P21 |
| *Cell dimensions* | | | |
| a, b, c (Å) | 95.01 159.67 47.31 | 86.90 165.54 47.64 | 37.74 136.05 38.38 |
| α, β, γ (°) | 90.00 90.00 90.00 | 90.00 90.00 90.00 | 90.00 109.12 90.00 |
| Resolution** (Å) | 47.31–2.73 (2.86–2.73) | 47.64–2.59 (2.60–2.59) | 68.03–1.82 (1.86–1.82) |
| $R_{sym}$ | 0.115 (1.940) | 0.073 (1.033) | 0.106 (0.950) |
| $R_{meas}$ | 0.119 (1.986) | 0.086 (1.216) | 0.115 (1.040) |
| $R_{pim}$ | 0.032 (0.557) | 0.045 (0.636) | 0.043 (0.416) |
| Total reflections | 136662 (16819) | 153763 (1503) | 222797 (9409) |
| No. unique reflections | 9913 (1279) | 41929 (417) | 31882 (1535) |
| CC1/2 | 0.999 (0.894) | 0.999 (0.447) | 0.995 (0.884) |
| $I/\sigma(I)$ | 22.6 (1.9) | 13.2 (1.0) | 11.8 (1.7) |
| Completeness (%) | 99.6 (99.5) | 99.9 (100.0) | 98.4 (96.4) |
| Multiplicity | 13.8 (13.2) | 3.7 (3.6) | 7.0 (6.1) |
| Wilson B-factor | 62.10 | 54.33 | 25.21 |
| *Refinement* | | | |
| Resolution (Å) | 40.82–2.73 | 47.64–2.59 | 35.66–1.82 |
| No. unique reflections | 9746 (890) | 41726 (4193) | 31170 (3109) |
| $R_{work}$ | 0.2675 (0.3751) | 0.2216 (0.2838) | 0.1834 (0.2821) |
| $R_{free}$ | 0.2779 (0.4246) | 0.2553 (0.3566) | 0.2402 (0.3060) |
| *R.m.s.d values* | | | |
| Bond lengths (Å) | 0.003 | 0.004 | 0.006 |
| Bond angles (°) | 0.462 | 0.696 | 1.159 |
| *No. of atoms* | | | |
| Macromolecule | 1744 | 7145 | 2707 |
| Ligands | 48 | 75 | – |
| solvent | 27 | 311 | 366 |
| *B-factors (Å²)* | | | |
| Protein | 85.02 | 70.59 | 38.31 |
| Ligand/ions | 81.89 | 56.10 | – |
| solvent | 57.15 | 48.83 | 40.80 |
| *Ramachandran plot* | | | |
| Favored (%) | 97.4 | 97.4 | 99.0 |
| Allowed (%) | 2.6 | 2.6 | 1.0 |
| Outliers (%) | 0.0 | 0.0 | 0.0 |
| Clashscore | 15.4 | 8.26 | 3.97 |
| Rotamer outliers (%) | 0.0 | 1.0 | 0.0 |
| *PDB code* | 9EFD | 9EFE | 9EFF |

* Twinned crystal. Estimated twinning fraction 0.48 for -h,-k,l.
** Highest-resolution shell values are shown in parentheses.

and purification of UBP mutants were performed as that of the UBP wild-type.

## Structure Determination

UBP (SIRe_1573) (7–8 mg/ml) crystals grew by mixing 1:1 v/v of protein and mother liquor containing Sodium Acetate 0.1 M pH 4.8–5.2, Lithium Chloride 0.2 M (or Magnesium Chloride 0.2 M), and PEG6000 18–24% at 20 °C using the sitting-drop vapor-diffusion method.

Crystals were harvested, cryo-protected in reservoir solution supplemented with 20% glycerol and flash-frozen in liquid nitrogen. UBP and DNA were mixed in buffer Tris 20 mM and NaCl 100 mM in a molar ratio of 1:1.2 to a final protein concentration of 4–5 mg/ml. The DNA duplex was prepared by annealing P49 and P50 (Table S1) in equal molar concentration. Crystals grew mixing 1:1 v/v of protein and mother liquor containing Tris/HCl 0.05 M pH 8.0–8.5, Potassium chloride 0.1 M, Magnesium chloride 0.01 M and PEG400 26–30% at 20 °C. Crystals were cryo-protected in the same reservoir solution supplemented with extra 10–15% of PEG400. Diffraction data were collected at 100 K at the Beamline station 4.2.2 at the Advanced Light Source (Berkeley National Laboratory, CA) and were initially indexed, integrated, and scaled using XDS[46]. For UBP (SIRe_1573) molecular replacement was used to estimate phases using PHASER and a predicted model generated by Alphafold[47] as the search model. The DNA-bound form data was phased using Single Anomalous Diffraction (SAD) on PHENIX Autosol. Initially, four Bromine atoms were found with occupancies of 0.8–0.9. A Figure of Merit (FOM) of 0.46 was obtained, and an incomplete model with Rwork/Rfree of 0.434/0.458 was built. In each case, after phasing, successive cycles of automatic building in Autobuild (PHENIX) and manual building in Coot, as well as refinement (PHENIX Refine), led to models with excellent geometry. MolProbity software[48] was used to assess the geometric quality of the models. Data collection and refinement statistics are indicated in Table 1. Inter-protomer interface size was calculated using PDBePISA (https://www.ebi.ac.uk/pdbe/pisa/).

## MCM N-terminus cloning, expression and purification

The *Sulfolobus islandicus* REY15A, *mcm* (SIRe_1228) gene, a 798 bp DNA fragment of *mcm* gene, codes for (1–266) amino acids, MCM N-terminus (MCM-NT) domain, was amplified by primers (P3 and P4) (Table S1) and cloned into pET30a vector, for the expression of (32.1 kDa/274aa) MCM-NT fusion protein with C-terminal 6×-Histidine tag. BL21DE3 Rosetta *E. coli* cells transformed with this construct were grown and induced at $OD_{600nm} = 0.8–0.9$, with 1 mM IPTG, for 4 h. The purification process of MCM-NT was carried out as described[24].

## Electrophoretic mobility shift assay

EMSA was performed in a 10 µl reaction volume containing 50 mM Potassium acetate, 20 mM Tris-acetate, 10 mM Magnesium acetate, 1 mM DTT, pH 7.9, supplemented with 5 or 10% glycerol, 70 mM NaCl and 800 ng salmon-sperm DNA. Typically, 10 ng of ds or ss-DNA substrates (P5–P18) with in defined lengths contains *oriC1* DUE sequence containing UCM motifs were incubated with the indicated amount of UBP at room temperature for 10 min before subjecting to electrophoresis (Table S1). The binding assay was performed at elevated temperatures (55 °C, 60 °C, 65 °C) in 20 µl reaction buffer comprised of 20 mM Tris pH 8.0 with 50 mM NaCl, with 2 nM DNA substrates (P19 and P20; predicted $T_m$ = 50.4 °C). Reactions were first incubated at room temperature for 10 min followed by 60 min at elevated temperatures. The EMSA assay to test for selective interaction of full-length MCM or MCM N-terminal domain with *oriC1*-bound UBP complex over the free DNA substrate was carried out with 11 nM of 140 bp DNA substrate (P21 and P22) containing *oriC1* sequence (Table S1).

The 5′-end radiolabelled of DNA substrates were carried out on 100 ng of either top or bottom strand with 30 µCi of 6000 Ci/mmol ATP (γ–$^{32}$P) (Revvity) using T4-Polynucleotide Kinase (New England Biolabs) and then annealed with its complementary strand. The protein-bound and unbound DNA substrates were resolved by 8% or 6% Native PAGE in Tris glycine buffer. Gel electrophoresis was carried out at 200 V for 80 minutes then gel dried and phosphorimagery performed (Typhoon biomolecular imager, Cytiva). The phosphor-image signals were quantified by ImageJ and graphs were plotted based on percentage values of the signals.

## DNase I footprinting

DNase I footprinting of UBP and Orc1-1 with *oriC1*: The EcoRI and NotI restriction digest fragments of the *oriC1* containing pPCR-Script plasmid, 181 bp DNA substrate fragment (P23 & P24) contains (ORB2-DUE-ORB3) sequence motifs (Table S1). The selective 3'-end labeling at EcoRI site of the bottom-strand was performed with 40 μCi of 3000 Ci/mmol dATP (α-$^{32}$P) (Revvity) using Klenow enzyme (New England Biolabs) as described[15].

DNase I foot printing was carried out with the indicated amount of UBP and Orc1–1 with 0.85 nM DNA substrate in 10 μl reaction containing 50 mM Potassium acetate, 20 mM Tris-acetate, 10 mM Magnesium acetate, 1 mM DTT, pH 7.9, supplemented with 4% glycerol, 70 mM NaCl and 8000 ng salmon-sperm DNA. Following the 10 min of binding reaction at room temperature, treated with 1 μl containing 0.4 units of DNase I (New England Biolabs) for 2 min and the enzymatic digestion was terminated with 10 μl loading buffer containing 1× TBE and Bromophenol dye. The samples were boiled at 95 °C for 5 min before subjecting to electrophoresis on a 6% PAGE containing 8 M Urea and 1× TBE. Maxam Gilbert A + G sequencing ladder was prepared and included beside the footprint sample lanes as described[15]. The same reaction buffer condition was used in the DNase I footprint on annealed oligos P5 and P6 (UCM motif in the DUE sequence of *oriC1*) with the indicated amount of UBP with both 5'-end radiolabelled top and bottom strand with ATP (γ-$^{32}$P) with enzymatic activity of T4-PNK (Table S1). 0.4 unit of DNase I was added to the reaction and incubated for 3 min. The defined strand length fragments (P25–P30) of (P5 and P6) were used as ruler (40, 30, and 20 nt) besides the DNase I test samples and were resolved on 12% PAGE containing 8 M Urea and 1× TBE at 60 W for 2 h (Table S1).

## Yeast 2 hybrid

The full-length open reading frame of UBP (SiRe_1573) (P39 and P40), Mcm (SiRe_1228) (P37 and P38), Orc1–1(SiRe_1740) (P35 and P36), MCM's N-Terminal domain (P41 and P42), AAA+ Core (P43 and P44), and HTH(P45 and P46) gene (P&P), MCM N-Terminal domain's-A (P41 and P47) and B/C(P48 and P42)-subdomains were PCR amplified by respective primers and cloned into respective double restriction-enzymes sites of the prey pGADT7 and bait pGBKT7 vector plasmid (Table S1). Yeast transformation and plating were performed as described in the Matchmaker protocol from the supplier (Clontech).

## Gel filtration chromatography

In total, 50 g (0.5 ml) of wild-type and mutant variants of UBP were applied to a Superdex 75 Increase 10/300 GL (GE Healthcare) column to profile protein oligomeric status. The protein oligomers were separated with a flow rate of 0.5 ml/min of TBS (10 mM Tris pH 8.0, 150 mM NaCl) buffer, and their elution was collected in fractions of 1 ml and confirmed by 15% SDS-PAGE. The corresponding UV (280 nM) absorbance elution peak profiles were plotted in the graph as a function of their elution volume.

## Pulldown assay

GST pulldown assay was similarly performed as described[49]. In total, 36 μg wild-type UBP protein was used as prey to perform the binding interaction with the indicated bait GST-fusion proteins prebound to Glutathione Sepharose beads. Western blotting was performed to confirm the protein-protein interaction.

## Estimation of cellular molar concentration of UBP

The molar quantity of UBP per *S. islandicus* REY15A cells was estimated based on the correlation of western signals from the cell extract of a defined number of cells with the titration range of purified recombinant UBP. The western signals were quantified by ImageJ, and a calibration curve of the titration of recombination UBP was plotted based on the percent value of the western signal. The western signal from the cell extract of defined cells, which is within the middle of the range of the titration series of recombinant protein, was used to calculate equivalent protein concentration.

## UBP expression and growth curve

*S. islandicus* E233S (S. islandicus REY15A Δ*pyrEF*Δ*lacS*Δ*argD*) was grown in 1×-TSVY growth medium with Uracil (20 μg/mL). The growth curve was monitored from early log Phase O.D$_{600}$ = 0.2 for 72 h duration and samples were collected at regular intervals. The western blot, probing for UBP or TBP as a loading control, was performed to confirm the UBP expression levels during the growth curve.

## UBP overexpression

*S. islandicus* REY15A (S. islandicus REY15A Δ*pyrEF*Δ*lacS*Δ*argD*) cells transformed with pSSRgD (empty vector) and pSSRgD-UBP were grown up to early log phase O.D$_{600}$ = 0.2 and resuspended in fresh media containing 0.2% D-(-)-Arabinose in 1×-TSVY media with with resulting O.D$_{600}$ = 0.1. The samples collected at regular intervals were analyzed to confirm the overexpression of UBP by Western blotting and RNA-Seq.

## Quantitative PCR

SYBR detection assays were performed in triplicate using 1 μl of ChIP DNA as template, forward and reverse primers at 125 nM final concentration, and Luna Universal qPCR Master Mix (New England Biolabs) according to the manufacturer's instructions. Standard curves were prepared in duplicate using ChIP input DNA as a template. qPCR reactions and melting curves were performed and analyzed using a Mastercycler ep realplex$^2$ machine and included analysis software (Eppendorf).

## MCM recruitment assay

These were performed as described previously[12]. Briefly, substrate DNA was prepared as described[14]. Binding reactions contained 200 pM DNA in 50 μl of binding buffer (20 mM Tris acetate, 50 mM potassium acetate, 10 mM magnesium acetate, 1 mM DTT, pH 7.9) plus 20 ng/μl poly dG:dC. Reactions were pre-heated to 50 °C in a water bath, and UBP and/or Orc1-1 protein (1 μM final) were added and supplemented with 2 mM ATP. Following a 10 min incubation at 50 °C, 50 μl of recombinant MCM [12.9 nM] plus competitor DNA and 2 mM ATP were added to the Orc1–1 reactions and incubated for 45 min at 50 °C. The reactions were then washed twice by magnet-mediated bead pelleting and resuspension in 100 μl binding buffer, followed by one wash in 100 μl binding buffer containing 500 mM potassium acetate. Following this final wash, beads were boiled in 1× SDS-PAGE loading buffer, run on SDS-PAGE gels, transferred to PVDF membranes and subjected to western analyses. All wash steps employed pre-warmed buffers and were performed with the tubes immersed in a 50 °C water bath.

## Statistics and reproducibility

Statistical approaches are outlined in the legends to figures and tables. Experiments were performed at least twice.

## Reporting summary

Further information on research design is available in the Nature Portfolio Reporting Summary linked to this article.

# Data availability

All NGS data have been deposited in the Sequence Read Archive, submission SUB15213019. Structural coordinates are deposited at the Protein Data Bank (PDB) under accession numbers 9EFD, 9EFE, and 9EFF. Requests for other material should be addressed to SDB. Source data are provided with this paper Source data are provided with this paper.

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

## Acknowledgements

This work was supported by grants to SDB from the Wellcome Trust (Grant 086045/Z/08/Z) and the National Institutes of Health (Grants R01GM135178 and R35GM152171). The authors gratefully acknowledge use of the Macromolecular Crystallography Facility (MCF) in the Molecular and Cellular Biochemistry Department, Indiana University Bloomington. We also thank Jay Nix for his assistance during X-ray data collection at beamline 4.2.2 at Advance Light Source (ALS), Berkeley, CA.

## Author contributions

R.D., R.Y.S., X.F., A.C., and G.G.-G. performed experiments, S.D.B supervised the experimental work and helped analyze data. S.D.B. wrote the initial draft of the paper. All authors commented on the paper. S.D.B. obtained funding for the work.

## Competing interests

The authors declare no competing interests.
