## [Transparent Peer Review file · Nature Communications]

An archaeal nucleoid-associated protein binds an essential motif in DNA replication origins

Corresponding Author: Professor Stephen Bell

Version 0:

Reviewer comments:

Reviewer #1

(Remarks to the Author)

This manuscript focuses on DNA replication initiation in archaea (*Sulfolobus islandicus* REY15A). It shows that the ucm sequence element within a replication origin (oriC1) is essential for origin activation. It also shows that the ucm element is the specific binding site for an essential protein UBP. The role of UBP is characterized in vivo and in vitro, leading to a model for UBP function at an archaeal replication origin.

Points to Consider:

1. Line 77. Replace "likely" with "potential".
2. Line 148. Replace "no" with "minimal".
3. Figure 2e. Is the ability to stabilize a dsDNA substrate specific to UBP? Or is this a general property of dsDNA binding proteins?
4. Line 182. Replace "no" with "minimal".
5. Line 192. Can the interaction of UBP with a 12 bp dsDNA substrate (i.e. same as used for crystallography) be detected using EMSA?
6. Comparing the footprint in Figure 2f with the x-ray structure of UBP bound to dsDNA, are these results compatible? The footprint seems to be larger than 12 bp, possibly displaying a region of strong protection and a region of weak protection. Is this a consequence of using DNase I rather than a smaller chemical cleavage method?
7. Line 220. Insert a period after "MCM".
8. Line 226. Delete "in the" for clarity (correct meaning?).
9. Regarding the protein:protein interaction between UBP and MCM, could this be modelled using AlphaFold?
10. Line 273. Are the ucm sites located within oriC2 and oriC3 also essential for the activity of the respective origins?
11. Line 289. Add "were observed" after "time points".
12. Line 290. Could the proposed enhanced cell growth be supported by plating cultures and enumerating colony forming units? Could the change in optical density arise from a change in cell shape or size?
13. Line 314. Remove the period after "Walker".
14. Figure 7a. Is the structure of the high salt stable complex known? Whether known or hypothetical, indicate the high salt and low salt complexes within the pathway shown in Figure 7b.
15. Figure 7a. Can this assay be modified to test the model that MCM loading leads to displacement of UBP from DNA (i.e. immunoblot for UBP)?
16. Figure 7b. Add arrows to indicate the directionality of the pathway.
17. What is the evidence to support the model that MCM loading leads to displacement of UBP from DNA? Is it the anticorrelation of UBP and MCM enrichment at oriC1 in the ucm mutant (M7) (Figure 5c)?
18. Figure 7b and the model. Should UBP display asymmetry in the illustration? Could such asymmetry play a role in either coordinated or sequential activation of MCM?
19. The mechanism for how UBP activates oriC1 remains unclear. Alternative hypotheses could be presented in the Discussion. For example, it appears that the Alba protein can bind to the ucm sequence (albeit non-specifically), so perhaps other NAPs can as well. Therefore, could an alternative model be that UBP occludes the DUE from other NAPs, which if allowed to bind inhibit origin activation?
20. Methods. Standardize concentration nomenclature throughout (either with or without a space between number and measurement).
21. Line 599. Delete "in".

22. Line 605. Was this 5% glycerol in addition to the 5-10% glycerol within the reaction mixture (lines 597-598)?

23. Figure 2 legend. Please provide lengths of various DNA substrates used in this figure.

24. Figure 2f legend. Are these two panels for the two strands (labelled in separate reactions)?

25. Line 963. Capitalize "In".

Reviewer #2

(Remarks to the Author)

Loading of replicative helicases onto DNA is fundamental of life. Despite recent advances in understanding the molecular details for helicase loading in bacteria and eukaryote, the underlying mechanisms in archaea remain obscure. In this study, Dhanaraju et al. combined biochemical and structural analyses to demonstrate that UBP, a previously uncharacterized DNA binding protein, plays a role in DNA replication. UBP binds to the origin of replication in a sequence-specific manner, and abolishing this binding site compromises origin firing *in vivo*, supporting the idea that UBP participates in initiation of DNA replication. Genome-wide analyses further suggested UBP may function as a global DNA binding protein, rather than an origin-specific replication initiation factor.

While the experiments are well conducted, the role of UBP in replication initiation is not demonstrated with sufficient clarity. The authors rely on the phenotype of origin mutants M6 and M7 as the evidence for UBP's essential role in initiation. However, they do not address the possibility that these mutants could affect Orc binding or the functions of other replication proteins—whether known or unknown—*independent* of UBP. Supporting this interpretation, the authors' *in vitro* assays suggest that UBP has little or no role in MCM loading. Given the global distribution of UBP across the chromosome, it remains plausible that UBP plays an indirect role in initiation.

To substantiate the conclusion that UBP plays an essential role in initiation, it is crucial for the authors to provide direct evidence showing that UBP promotes the initiation process *in vitro*.

Detailed comments are outlined below:

P1, title: Please consider rephrasing the title. As the current data do not provide direct evidence for the essentiality of UBP in DNA replication, the title may overstate the conclusions drawn in the study.

P2, 26-28: As mentioned above, the current data are not sufficient to support the authors' view. It remains unclear whether UBP plays a key role in helicase loading.

P2, 115-116, Figure 1b: Seemingly, M8 had the inhibitory effects on the *oriC2* activity. Is this reproducible?

P5, 121-122, Figure 1c: Ideally, a western blot for a strain lacking *Orc1* should be included as a negative control. If the specificity of the antibodies has already been validated in previous work, please cite the relevant references.

P6, 147-148, Figure 2d: Please quantify the data presented in Figure 2d. It appears that mutant M6 largely retains its binding affinity. If the authors claim that this mutant exhibits a loss-of-function phenotype, statistical analysis is required to substantiate this conclusion.

P8, 212-215: Figure 4a does not convincingly support the interaction between MCM and UBP. The authors may consider testing additional reporters to strengthen the evidence for this interaction. Additionally, protein expression levels should be assessed to compare the truncated versions shown in Figures 4c and 4d.

P8 228-231: The authors may consider to use DNA footprinting to investigate interaction between MCM and UBP on DNA.

P9 253-255: While the data are consistent, they do not provide evidence to support the essential role of UBP.

P10, 280-281: Please include details in the text regarding the amount of protein expressed: was it 10-fold, 100-fold, or more? The critical question is whether the observations under UBP overexpression is physiologically relevant or merely artefactual. It is plausible that overexpressed UBP disrupted normal cellular processes by binding nonspecifically across the chromosome, thereby inhibiting the functions of other DNA binding proteins such as *Orc* and transcription factors. The authors could address this concern by performing and presenting UBP-ChIP-seq data under conditions of UBP overexpression.

P11 303: change "UBP-per" to "UBP-over".

Did the authors measure the levels of MCM under UBP overexpression?

P11 314: change "Walker. B" to "Walker B".

P12-13: The authors may consider restructuring the Discussion section. In particular, the latter half of the first paragraph is challenging to follow and does not clearly convey the authors' main message. This part places excessive emphasis on existing literature regarding the molecular mechanisms of MCM helicase loading, without adequately connecting it to the current findings on UBP. A clearer focus on how these findings contribute to or contrast with prior knowledge would enhance the Discussion.

P30: the legend for Figure 5 is missing.

Figure 5b: The UBP protein sizes appeared to differ between 0-2h and 4-24h time points. Could this indicate partial degradation of UBP?

Figure 5c: Is *Orc* binding affected in the M7 mutant?

Figure 5g: Out of curiosity, the consensus sequence of UBP appears to be nearly identical to *DnaA* box. This could be coincidental, but I wonder if the authors have any thoughts or insights about this similarity.

Figure 6b: Please provide a detailed explanation of the DNA profiles observed in the flow cytometry data. The peak at the origins in Figure 1b suggests a mixed cell-cycle population in wild-type cells. Why, then, do most cells in Figure 6b appear to contain two chromosomes?

Reviewer #3

(Remarks to the Author)

One fundamental challenge of cells is to accurately copy their genetic material for cell proliferation. All organisms in the three domains of life carry out semiconservative DNA replication as originally hypothesized by Watson and Crick. DNA replication is performed by sophisticated multi-protein machines, which must synthesize DNA continuously and copy it accurately at the proper time in the cell cycle. The core replication proteins present in all cell types include a helicase that separates the DNA duplex. Chromatin or nucleoid-associated proteins have been shown to modulate origin usage in eukaryotes and bacteria.

The current ms by Dhanaraju and colleagues reports the identification and characterization of an archaeal nucleoid-associated protein (UBP) involved in DNA replication initiation. The authors report the crystal structure of UBP, both in its apo and DNA-bound forms, and show that it interacts with the N-terminal region of MCM (the main replicative helicase in both eukaryotes and archaea). The authors also investigated the sequence elements required for in vivo function of the DNA replication origin oriC1 in *Sulfolobus islandicus* REY15A.

This impressive work provides new insights into the mechanism of origin activation in archaea and will be of high interest to a broad audience.

Below, I have listed several suggestions that may help the authors improve their manuscript:

1) Page 6, line 167: The following sentence seems confusing and should be revised: "When viewed from underneath the stirrups the proteins adopt a rising left-handed spiral" Also, please define what is the stirrup loop (amino-acid range and topological location in the structure).

2) Page 7, lines 168-169: The authors may consider quantifying the homodimerization surface (buried surface area) using the EMBL-PISA server for example.

3) Figure 3 requires more extensive annotations :

- Panel 3a: Consider including labels for the N- and C-terminal ends of the two chains, numbering the secondary structure elements (e.g., $\alpha 1$, $\alpha 2$, $\beta 1$, $\beta 2$, etc.), and highlighting the stirrup loop.

- Panel 3b: Please add a Coulombic potential scale bar.

- Panel 3c: Consider coloring the atoms by chemical element to help readers appreciate the nature of the interactions.

Highlighting critical residues and interactions may also help.

- Panel 3d: Please indicate the 3' and 5' ends of the two DNA strands, as well as the location of the stirrup loop.

- Panel 3e: Is the DNA orientation consistent with panel 3d? Ensuring this alignment could help readers better orient themselves within the structure.

4) Figure 3b: the surface electrostatic potential of the UBP dimer suggests that it could bind to DNA (possibly encircle the DNA). However, the EMSAs show a single band for the UBP-DNA complexes. This might be worth discussing in the ms.

5) Lines 215-218: Experiments shown in figure 4 show that UBP interacts with both A and B/C domains. This suggests that UBP and MCM interact together through a rather wide interaction surface. Could such interface be predicted using Alpha-Fold? Could the authors comment on the nature of the interaction, stable or transient?

6) Please edit the typos lines 113 & 889

Version 1:

Reviewer comments:

Reviewer #1

(Remarks to the Author)

The authors have satisfactorily addressed all points raised in this review.

Reviewer #2

(Remarks to the Author)

I am mostly satisfied with the authors' responses but still have some concerns regarding the EMSA data in Figure 2. I do not dispute the appearance of the band intensities and acknowledge the authors' point that even a partial reduction in DNA binding could have a significant impact in vivo. However, given that quantification and statistical analysis of band intensities are widely accepted practices, incorporating these analyses would further strengthen the validity of the conclusions.

We thank the reviewers for their helpful and constructive critiques. As outlined below, we have made modifications to the manuscript in line with their recommendations.

REVIEWER COMMENTS

Reviewer #1 (Remarks to the Author):

This manuscript focuses on DNA replication initiation in archaea (*Sulfolobus islandicus* REY15A). It shows that the ucm sequence element within a replication origin (oriC1) is essential for origin activation. It also shows that the ucm element is the specific binding site for an essential protein UBP. The role of UBP is characterized in vivo and in vitro, leading to a model for UBP function at an archaeal replication origin.

Points to Consider:

1. Line 77. Replace “likely” with “potential”.

Done

2. Line 148. Replace “no” with “minimal”.

Done

3. Figure 2e. Is the ability to stabilize a dsDNA substrate specific to UBP? Or is this a general property of dsDNA binding proteins?

There is precedent for other dsDNA binding proteins stabilizing duplex DNA (e.g. Sac7d and Alba in archaea). We were keen to include this experiment in light of the DUE-binding of UBP. Entering into this work, one possible mode of action of the protein that we considered was that it might aid duplex melting at this crucial site in the origin. Our EMSAs indicate that it does not stably bind single-stranded DNA but we wished to test whether it might be able to effect melting. Our data suggests it does not.

4. Line 182. Replace “no” with “minimal”.

Done

5. Line 192. Can the interaction of UBP with a 12 bp dsDNA substrate (i.e. same as used for crystallography) be detected using EMSA?

Yes, we have included this data as a new Supplementary Figure 4a . We have also included a figure (Supplementary Fig 4b) confirming the BrdU substitutions to the oligo (necessary for solving the phasing) do not affect binding.

6. Comparing the footprint in Figure 2f with the x-ray structure of UBP bound to dsDNA,

are these results compatible? The footprint seems to be larger than 12 bp, possibly displaying a region of strong protection and a region of weak protection. Is this a consequence of using DNase I rather than a smaller chemical cleavage method?

Yes, we believe so.

7. Line 220. Insert a period after “MCM”.

Done.

8. Line 226. Delete “in the” for clarity (correct meaning?).

Thanks – this has been corrected.

9. Regarding the protein:protein interaction between UBP and MCM, could this be modelled using AlphaFold?

We have tried to generate meaningful models with AlphaFold3. However, we end up with 5 distinct models with UBP at various positions on the MCM. We have attempted with and without DNA and with monomer or hexameric MCM. We will note that AlphaFold fails to generate the crystallographically defined UBP dimer conformation and does not dock a monomer of UBP on the correct DNA sequence (if supplied with an extended *oriC1* DNA sequence that could accommodate MCM). With MCM, it consistently returns one conformation of the N-terminal domains of the protein relative to the B/C domains. Work from Enemark and colleagues (Miller, eLife, 2014 <https://doi.org/10.7554/eLife.03433>) has revealed that this region of archaeal MCM can adopt two distinct conformations based on rotation in the linker between A and B/C domains. Importantly, it has been proposed that this rotation could be linked to the replication initiation phenotype of the *mcm5-bob1* mutation in yeast. In light of these discrepancies between AlphaFold's predictions and experimentally determined structures, we would prefer not to include the AlphaFold predictions in the current manuscript. We are currently attempting to solve the cryo-EM structure of MCM-UBP-DNA but we hope the referee agrees that this goes beyond the scope of the current manuscript.

10. Line 273. Are the *ucm* sites located within *oriC2* and *oriC3* also essential for the activity of the respective origins?

We do not know at this time. We note that UBP does bind to these sequences *in vitro* (unpublished data) and *in vivo* (our ChIP-Seq data).

11. Line 289. Add “were observed” after “time points”.

Added

12. Line 290. Could the proposed enhanced cell growth be supported by plating cultures and enumerating colony forming units? Could the change in optical density arise from a change in cell shape or size?

We have included forward scatter data from the flow-cytometry measurements (new Supplementary Figure 6C). These indicate that the UBP-over-expressing cells are larger than the empty-vector counterparts.

13. Line 314. Remove the period after “Walker”.

Done

14. Figure 7a. Is the structure of the high salt stable complex known? Whether known or hypothetical, indicate the high salt and low salt complexes within the pathway shown in Figure 7b.

We are currently attempting to characterize the structures of the different high and low salt stable complex. We have expanded our description of the nature of these forms of MCM and modified Figure 7b to facilitate interpretation of the data.

15. Figure 7a. Can this assay be modified to test the model that MCM loading leads to displacement of UBP from DNA (i.e. immunoblot for UBP)?

Despite several years of effort we do not (frustratingly!!) yet have an in vitro assay for origin melting or origin-mediated DNA replication. So, while the current assay gives an indication of the relative abilities of Orc1-1 and UBP to recruit MCM to the origin-containing DNA, we do not think it provides information on downstream events. Also, our estimates based on our previous work indicate that only maximally 20% of Orc1-1-bound DNA loads MCM in these assays. We are currently attempting to establish ensemble and single molecule assays to look at the downstream events.

16. Figure 7b. Add arrows to indicate the directionality of the pathway.

Done

17. What is the evidence to support the model that MCM loading leads to displacement of UBP from DNA? Is it the anticorrelation of UBP and MCM enrichment at oriC1 in the *ucm* mutant (M7) (Figure 5c)?

At this point our proposal for the mechanism for displacement of UBP is speculative. However, the DUE region, and thus the *ucm*, will melt during replication initiation and we do not detect single-stranded DNA binding by UBP. Thus, UBP must be displaced at

some point in the process. As alluded to our response to point 15, we are currently attempting to establish single-molecular assays to determine this.

18. Figure 7b and the model. Should UBP display asymmetry in the illustration? Could such asymmetry play a role in either coordinated or sequential activation of MCM?

This is a great point, we have expanded on the issue of symmetry in the discussion. Since we have not yet mapped the surface of UBO responsible for interaction, we would prefer to retain a featureless circle to indicate a monomer of UBP in the cartoon model.

19. The mechanism for how UBP activates oriC1 remains unclear. Alternative hypotheses could be presented in the Discussion. For example, it appears that the Alba protein can bind to the *ucm* sequence (albeit non-specifically), so perhaps other NAPs can as well. Therefore, could an alternative model be that UBP occludes the DUE from other NAPs, which if allowed to bind inhibit origin activation?

Absolutely, we have added some discussion of alternative models. We will note that, with respect to the suggestion that UBP is excluding other NAPs it is hard to reconcile this proposal with our observation that MCM occupancy of the origin is enhanced upon mutation of the *ucm*.

20. Methods. Standardize concentration nomenclature throughout (either with or without a space between number and measurement).

Done

21. Line 599. Delete “in”.

Done

22. Line 605. Was this 5% glycerol in addition to the 5-10% glycerol within the reaction mixture (lines 597-598)?

Ah, our apologies, final concentration of glycerol prior to gel loading was 5 – 10 %.

23. Figure 2 legend. Please provide lengths of various DNA substrates used in this figure.

Done

24. Figure 2f legend. Are these two panels for the two strands (labelled in separate reactions)?

The referee is correct, we have added this information to the figure legend.

25. Line 963. Capitalize “In”.

Done

Reviewer #2 (Remarks to the Author):

Loading of replicative helicases onto DNA is fundamental of life. Despite recent advances in understanding the molecular details for helicase loading in bacteria and eukaryote, the underlying mechanisms in archaea remain obscure. In this study, Dhanaraju et al. combined biochemical and structural analyses to demonstrate that UBP, a previously uncharacterized DNA binding protein, plays a role in DNA replication. UBP binds to the origin of replication in a sequence-specific manner, and abolishing this binding site compromises origin firing *in vivo*, supporting the idea that UBP participates in initiation of DNA replication. Genome-wide analyses further suggested UBP may function as a global DNA binding protein, rather than an origin-specific replication initiation factor.

While the experiments are well conducted, the role of UBP in replication initiation is not demonstrated with sufficient clarity. The authors rely on the phenotype of origin mutants M6 and M7 as the evidence for UBP’s essential role in initiation. However, they do not address the possibility that these mutants could affect Orc binding or the functions of other replication proteins— whether known or unknown— independent of UBP. Supporting this interpretation, the authors’ *in vitro* assays suggest that UBP has little or no role in MCM loading. Given the global distribution of UBP across the chromosome, it remains plausible that UBP plays an indirect role in initiation.

To substantiate the conclusion that UBP plays an essential role in initiation, it is crucial for the authors to provide direct evidence showing that UBP promotes the initiation process *in vitro*.

Detailed comments are outlined below:

P1, title: Please consider rephrasing the title. As the current data do not provide direct evidence for the essentiality of UBP in DNA replication, the title may overstate the conclusions drawn in the study.

We have changed the title to

Identification and characterization of an archaeal nucleoid-associated protein that binds an essential motif in DNA replication origins.

P2, 26-28: As mentioned above, the current data are not sufficient to support the authors’ view. It remains unclear whether UBP plays a key role in helicase loading.

With respect, at no point in this manuscript do we propose that UBP plays a key role in helicase loading - our *in vitro* data shows it plays a minor role (compared to Orc1-1) in

recruitment and our *in vivo* data reveal that upon mutation of UBP's binding site, MCM actually shows elevated signal at the replication origin. As we discuss extensively in the manuscript, our data therefore suggest the primary function is in helicase egress from the origin, not recruitment.

P2, 115-116, Figure 1b: Seemingly, M8 had the inhibitory effects on the *oriC2* activity. Is this reproducible?

We consistently see noisy MFA data with the M8 strain, we are not sure at this time why this is the case. However, it is clear that *oriC1* is still firing.

P5, 121-122, Figure 1c: Ideally, a western blot for a strain lacking Orc1 should be included as a negative control. If the specificity of the antibodies has already been validated in previous work, please cite the relevant references.

We published this in 2013 (Samson et al, Cell Reports, figure reproduced below)

Figure 1. Individual Orc1/Cdc6 Genes Are Nonessential
(A) Western analyses of Orc1/Cdc6 levels in wild-type (wt) and strains deleted for *orc1-1* ($\Delta 1$), *orc1-2* ($\Delta 2$), or *orc1-3* ($\Delta 3$). Antiserum against PCNA3 was used as a loading control.

P6, 147-148, Figure 2d: Please quantify the data presented in Figure 2d. It appears that mutant M6 largely retains its binding affinity. If the authors claim that this mutant exhibits a loss-of-function phenotype, statistical analysis is required to substantiate this conclusion.

Respectfully, we disagree, the yield of stable complex for UBP on M6 is clearly lower than seen with the wild-type, indicating reduced affinity. Furthermore, the referee should bear in mind that *in vivo* hundreds of additional sites on the chromosome will be competing for UBP binding, even a modestly lowered affinity *in vitro* could have a far more significant impact in the *in vivo* context.

P8, 212-215: Figure 4a does not convincingly support the interaction between MCM and UBP. The authors may consider testing additional reporters to strengthen the evidence for this interaction. Additionally, protein expression levels should be assessed to compare the truncated versions shown in Figures 4c and 4d.

We agree that the 2 hybrids suggest a weak interaction, that's why we additionally performed the in vitro pull-down and EMSA supershift assays to corroborate the interaction. Taken together the experiments in Figure 4 support an interaction between MCM.

P8 228-231: The authors may consider to use DNA footprinting to investigate interaction between MCM and UBP on DNA.

We did attempt this, however, as MCM readily binds double stranded linear DNA and completely protects it from digestion, the results were impossible to interpret.

P9 253-255: While the data are consistent, they do not provide evidence to support the essential role of UBP.

In light of the referee's comments, we have added the sentence

"Thus, these data suggest that UBP's role at the replication origin is not in recruitment of MCM but, rather, in facilitating MCM's exit from the origin."

P10, 280-281: Please include details in the text regarding the amount of protein expressed: was it 10-fold, 100-fold, or more? The critical question is whether the observations under UBP overexpression is physiologically relevant or merely artefactual. It is plausible that overexpressed UBP disrupted normal cellular processes by binding nonspecifically across the chromosome, thereby inhibiting the functions of other DNA binding proteins such as Orc and transcription factors. The authors could address this concern by performing and presenting UBP-ChIP-seq data under conditions of UBP overexpression.

The level of over-expression is shown in Supplementary Figure 5. We can include quantitation of these data in the manuscript.

P11 303: change "UBP-per" to "UBP-over".

Did the authors measure the levels of MCM under UBP overexpression?

We have not performed any measurement of MCM level in the UCM over-expressing strain.

P11 314: change "Walker. B" to "Walker B".

Done

P12-13: The authors may consider restructuring the Discussion section. In particular, the latter half of the first paragraph is challenging to follow and does not clearly convey the authors' main message. This part places excessive emphasis on existing literature regarding the molecular mechanisms of MCM helicase loading, without adequately connecting it to the current findings on UBP. A clearer focus on how these findings contribute to or contrast with prior knowledge would enhance the Discussion.

We have made some modifications to the discussion, in line with the referees's suggestions.

P30: the legend for Figure 5 is missing.

Added, our apologies for omitting this in the original version.

Figure 5b: The UBP protein sizes appeared to differ between 0-2h and 4-24h time points. Could this indicate partial degradation of UBP?

No, rather we believe the gel simply ran somewhat unevenly.

Figure 5c: Is Orc binding affected in the M7 mutant?

We have included a new panel in figure 5c (copy below) that confirms there is no significant alteration to Orc1-1 in vivo occupancy in M7 detectable in ChIP assays. In contrast the M12 mutation that impacts ORB2 shows reduced origin occupancy by Orc1-1.

C

Figure 5g: Out of curiosity, the consensus sequence of UBP appears to be nearly identical to DnaA box. This could be coincidental, but I wonder if the authors have any thoughts or insights about this similarity.

It's certainly intriguing and may reflect the eventual meltability of the sequence. We will emphasize that UBP itself neither melts DNA nor does it bind ssDNA of either strand containing this sequence.

Figure 6b: Please provide a detailed explanation of the DNA profiles observed in the flow cytometry data. The peak at the origins in Figure 1b suggests a mixed cell-cycle population in wild-type cells. Why, then, do most cells in Figure 6b appear to contain two chromosomes?

As described in our materials and methods, DNA for the MFA in Figure 1 were harvested at OD (600 nm) = 0.2. Our MFA assays indicate a ratio of less than 1.2 between origin peak and terminus trough. If all cells were undergoing DNA replication with no gap phases then the maximal ratio would be 2:1. The existence of extended gap phases (most significantly G2, as seen in our flow cytometry profiles) with cells that are not actively replicating results in a lower ratio.

The flow profile in WT cells harvested at 30 hours shows S-phase cells and G2 cells. This makes sense when one looks at the accompanying growth curve. At 30 hours, cells are at an OD of 1 and will reach stationary phase at 45 hours at an OD of 1.4 – less than one mass doubling. It is well-established that *Sulfolobus* cells in stationary phase have a 2-chromosome content. Thus, the 30 hour population will complete replication and growth will cease with very few cells undergoing cell division. In contrast the UBP overexpressing cells show ongoing active cell division at 30 hours (manifested

by the presence of the 1C population, in agreement with their extended window of growth).

Reviewer #3 (Remarks to the Author):

One fundamental challenge of cells is to accurately copy their genetic material for cell proliferation. All organisms in the three domains of life carry out semiconservative DNA replication as originally hypothesized by Watson and Crick. DNA replication is performed by sophisticated multi-protein machines, which must synthesize DNA continuously and copy it accurately at the proper time in the cell cycle. The core replication proteins present in all cell types include a helicase that separates the DNA duplex. Chromatin or nucleoid-associated proteins have been shown to modulate origin usage in eukaryotes and bacteria.

The current ms by Dhanaraju and colleagues reports the identification and characterization of an archaeal nucleoid-associated protein (UBP) involved in DNA replication initiation. The authors report the crystal structure of UBP, both in its apo and DNA-bound forms, and show that it interacts with the N-terminal region of MCM (the main replicative helicase in both eukaryotes and archaea). The authors also investigated the sequence elements required for in vivo function of the DNA replication origin oriC1 in *Sulfolobus islandicus* REY15A.

This impressive work provides new insights into the mechanism of origin activation in archaea and will be of high interest to a broad audience.

Below, I have listed several suggestions that may help the authors improve their manuscript:

1) Page 6, line 167: The following sentence seems confusing and should be revised: "When viewed from underneath the stirrups the proteins adopt a rising left-handed spiral" Also, please define what is the stirrup loop (amino-acid range and topological location in the structure).

Thanks for the suggestions, we've removed the sentence and modified Figure 3 and Supplementary Figure 2

2) Page7, lines 168-169: The authors may consider quantifying the homodimerization surface (buried surface area) using the EMBL-PISA server for example.

Great idea, thanks – PISA gives an inter-protomer interface of 1662.4 \AA^2 , we've added this information to the manuscript.

3) Figure 3 requires more extensive annotations :

- Panel 3a: Consider including labels for the N- and C-terminal ends of the two chains, numbering the secondary structure elements (e.g., $\alpha 1$, $\alpha 2$, $\beta 1$, $\beta 2$, etc.), and highlighting the stirrup loop.

Done, note that the labeling of secondary structure features is in a revised Supplementary Figure 2

- Panel 3b: Please add a Coulombic potential scale bar.

Done, my apologies for omitting this in the original version

- Panel 3c: Consider coloring the atoms by chemical element to help readers appreciate the nature of the interactions. Highlighting critical residues and interactions may also help.

Given the small size of this panel we felt adding additional labeling would overly clutter it. The key point for this panel is to illustrate that the electron density strongly supports homodimerization.

- Panel 3d: Please indicate the 3' and 5' ends of the two DNA strands, as well as the location of the stirrup loop.

Thanks for the suggestion, we've added these elements to the figure.

- Panel 3e: Is the DNA orientation consistent with panel 3d? Ensuring this alignment could help readers better orient themselves within the structure.

We've revised this panel in line with the referee's suggestion.

4) Figure 3b: the surface electrostatic potential of the UBP dimer suggests that it could bind to DNA (possibly encircle the DNA). However, the EMSAs show a single band for the UBP-DNA complexes. This might be worth discussing in the ms.

Thanks, we've expanded a bit on the dimer/monomer issue in the discussion

5) Lines 215-218: Experiments shown in figure 4 show that UBP interacts with both A and B/C domains. This suggests that UBP and MCM interact together through a rather wide interaction surface. Could such interface be predicted using Alpha-Fold? Could the authors comment on the nature of the interaction, stable or transient?

We have tried to generate meaningful models with AlphaFold3. However, we end up with 5 distinct models with UBP at various positions on the MCM. We have attempted with and without DNA and with monomer or hexameric MCM. We will note that AlphaFold fails to generate the crystallographically defined UBP dimer conformation and does not dock a monomer of UBP on the correct DNA sequence (if supplied with an extended *oriC1* DNA sequence that could accommodate MCM). With MCM, it consistently returns one conformation of the N-terminal domains of the protein relative to the B/C domains. Work from Enemark and colleagues (Miller, eLife, 2014

<https://doi.org/10.7554/eLife.03433>)has revealed that this region is of archaeal MCM can adopt two distinct conformations based on rotation in the linker between A and B/C domains. Importantly, it has been proposed that this rotation could be linked to the replication initiation phenotype of the *mcm5-bob1* mutation in yeast. In light of these discrepancies between AlphaFold's predictions and experimentally determined structures, we would prefer not to include the AlphaFold predictions in the current manuscript. We are currently attempting to solve the cryo-EM structure of MCM-UBP-DNA but we hope the referee agrees that this goes beyond the scope of the current manuscript.

6) Please edit the typos lines 113 & 889

Done

REVIEWERS' COMMENTS

Reviewer #1 (Remarks to the Author):

The authors have satisfactorily addressed all points raised in this review.

Great!

Reviewer #2 (Remarks to the Author):

I am mostly satisfied with the authors' responses but still have some concerns regarding the EMSA data in Figure 2. I do not dispute the appearance of the band intensities and acknowledge the authors' point that even a partial reduction in DNA binding could have a significant impact in vivo. However, given that quantification and statistical analysis of band intensities are widely accepted practices, incorporating these analyses would further strengthen the validity of the conclusions.

We've added the requested quantitation to the lower panel of Figure 2d.